# Prenatal alcohol exposure alters expression of genes involved in cell adhesion, immune response, and toxin metabolism in adolescent rat hippocampus

Amal Khalifa[1], Rebecca Palu[2]*, Amy E. Perkins[3]*, Avery Volz[2,3]

1 Department of Computer Science, Purdue University Fort Wayne, Fort Wayne, IN, United States of America, 2 Department of Biological Sciences, Purdue University Fort Wayne, Fort Wayne, IN, United States of America, 3 Department of Psychology, Purdue University Fort Wayne, Fort Wayne, IN, United States of America

⊛ These authors contributed equally to this work.
* palur@pfw.edu (RP); aeperkin@pfw.edu (AEP)

**Data Availability Statement:** RNA-seq data is available at NCBI GEO (accession number GSE247256). Additional relevant data are within the paper and its Supporting Information files.

## Abstract

Prenatal alcohol exposure (PAE) can result in mild to severe consequences for children throughout their lives, with this range of symptoms referred to as Fetal Alcohol Spectrum Disorders (FASD). These consequences are thought to be linked to changes in gene expression and transcriptional programming in the brain, but the identity of those changes, and how they persist into adolescence are unclear. In this study, we isolated RNA from the hippocampus of adolescent rats exposed to ethanol during prenatal development and compared gene expression to controls. Briefly, dams were either given free access to standard chow *ad libitum* (AD), pair-fed a liquid diet (PF) or were given a liquid diet with ethanol (6.7% ethanol, ET) throughout gestation (gestational day (GD) 0–20). All dams were given control diet *ad libitum* beginning on GD 20 and throughout parturition and lactation. Hippocampal tissue was collected from adolescent male and female offspring (postnatal day (PD) 35–36). Exposure to ethanol caused widespread downregulation of many genes as compared to control rats. Gene ontology analysis demonstrated that affected pathways included cell adhesion, toxin metabolism, and immune responses. Interestingly, these differences were not strongly affected by sex. Furthermore, these changes were consistent when comparing ethanol-exposed rats to pair-fed controls provided with a liquid diet and those fed ad libitum on a standard chow diet. We conclude from this study that changes in genetic architecture and the resulting neuronal connectivity after prenatal exposure to alcohol continue through adolescent development. Further research into the consequences of specific gene expression changes on neural and behavioral changes will be vital to our understanding of the FASD spectrum of diseases.

**Funding:** Funding provided by Purdue University Fort Wayne through a Collaborative Research Grant from Purdue University Fort Wayne (https://www. pfw.edu/offices/sponsored-programs/internal-grants-funding/collaborative-research-grant) to AK, RASP, and AP. The funders had no role in study design, data collection and analysis, decision to publish, or preparation of the manuscript.

**Competing interests:** The authors have declared that no competing interests exist.

## Introduction

Maternal consumption of alcohol is relatively common, with 13.5% of pregnant women reporting some drinking during pregnancy and 5.3% reporting binge drinking [1]. Given that alcohol easily crosses the placental barrier, maternal alcohol consumption can significantly affect fetal development. Prenatal alcohol exposure (PAE) is associated with a wide range of neurobehavioral outcomes, including altered sensory processing, intellectual disability, developmental delays, and cognitive dysfunction [2] PAE, combined with facial dysmorphology, impaired growth, and perturbations in behavior, may result in diagnosis of Fetal Alcohol Syndrome (FAS). FAS is part of the larger diagnostic umbrella of Fetal Alcohol Spectrum Disorders, which includes partial FAS and alcohol-related neurodevelopmental disorder [3]. Importantly, it is estimated that the prevalence of FASD in the United States is 1–5% [4].

Neuroimaging has been critical in elucidating the neuroanatomical and functional consequences of PAE. Not only is PAE correlated with overall reductions brain volume and major fiber tracts (e.g., corpus callosum), but there are specific reductions in the volume of subcortical structures such as the hippocampus [3, 5, 6]. Functional magnetic resonance imaging (fMRI) studies have reported altered neuronal activation during working memory and executive function tasks, but the data from neuroimaging studies has been inconsistent [7]. A recent study by Solar et al. (2022) used magnetic resonance imaging (MRI) and diffusion tensor imaging (DTI) to show that although the volume of the hippocampus was smaller in individuals with PAE, there were no significant group differences in mean diffusivity, axial diffusivity, or fractional anisotropy, suggesting that alcohol-related cognitive dysfunction is a result of changes at the cellular level [8]. Rodent models have been critical in evaluating the short- and long-term effects of PAE at the neuronal level. However, the consequences of developmental alcohol exposure are highly influenced by the timing of alcohol exposure (e.g,. prenatal vs. postnatal exposure), the dose of alcohol, the developmental epoch of measurement (e.g., early postnatal, adolescent, or adult), and brain region. For example, while prenatal alcohol exposure reduced the volume of the hippocampus at E17 and resulted in a reduction in NeuN + cells at P7 [9], there were no changes in cell proliferation in the adult hippocampus [10]. In addition, using unbiased stereology, Tran et al. (2003) found a significant reduction in neuron number in CA1 in adult rats following developmental alcohol exposure, but only when alcohol was delivered postnatally (P2-10) or pre- and postnatally (G1-22 + P2-10) [11]. Not only are there changes in the volume and/or numbers of hippocampal neurons following prenatal alcohol exposure, but there are also alterations in the neuroimmune system. For example, prenatal alcohol exposure produced elevated levels of several cytokines within the hippocampus at P8 [12] and altered neuroimmune response to challenge [13, 14]. Finally, developmental alcohol exposure alters hippocampal synaptic plasticity [15, 16], which may help to explain alcohol-related cognitive deficits. Together, these studies point to the hippocampus as being a key site of alcohol-related alterations.

Using transcriptomics and high-throughput genetic screening techniques, researchers have demonstrated that PAE significantly alters gene expression in humans and in animal models. For the most part, studies have shown that PAE results in a significant down-regulation of gene expression [17–20]. In a study evaluating DNA methylation following PAE, buccal samples were obtained from newborns and evaluated for differentially methylated regions, with few such regions identified [21]. However, others have looked at children and adolescents with FASD and reported significant differentially methylated regions. A study analyzing DNA methylation in buccal swabs (age range 3.5–18 years) reported a DNA methylation signature that may be unique to individuals with FASD [22]. Altered methylation of genes related to protocadherins was observed in buccal swabs of children with FASD [23]. Finally, DNA

methylation alterations were observed in blood samples in children and adolescents with FASD [24]. In addition, PAE alters gene expression in rodent fetal hippocampal tissue. Specific pathways identified as being altered by PAE include those involved in regulation of transcription [25], G-protein coupled receptor signaling [25, 26], proline and citrulline biosynthesis [27] and neuron differentiation [26].

However, few studies have utilized genome-wide screens to evaluate the long-term consequences of PAE. Of those that have, a limited number have focused on adolescence as a key period of interest. Furthermore, there is wide variation in the alcohol exposure paradigms used in previous studies. In the amygdala, PAE (gestational day (GD) 12) resulted in altered miRNA expression, with key pathways including those related to p53, CREB, glutamate and GABA [28]. Another study found that PAE (10% ethanol; GD 0.5–8.5) resulted in significant upregulation of genes involved in neuroinflammation in the hippocampus of male adolescent rats [29]. In the cortex, about 250 genes were downregulated as a result of PAE (5% ethanol liquid diet throughout gestation). Interestingly, far fewer genes were altered in female alcohol-exposed offspring relative to male alcohol-exposed offspring (37 vs. 271, respectively). Genes identified were associated with neurological disorders and the protocadherin family of proteins [19]. Finally, in a study comparing the effects of PAE (35% ethanol-derived calories GD 11–12) on gene expression in the olfactory bulb of adolescent (postnatal day (P) 40) and adult (P90) rats, fewer changes were evident in adolescents. In fact, only a single gene was identified as being upregulated in ethanol-treated rats compared to pair-fed controls, dual specificity phosphatase 1 (*Dusp1*), whereas no genes were downregulated [30]. Taken together, it is clear that PAE causes long-lasting alterations in gene expression, but these effects vary by brain region. Furthermore, some studies have evaluated sex differences [19, 28], whereas others have utilized only male rats [29] or collapsed across sex [30]. We conducted an exploratory study to examine whether PAE resulted in changes in gene expression in the adolescent male and female hippocampus using RNA sequencing. Based on previous research described above, we hypothesized that alcohol exposure would result in significant downregulation of gene expression, with highlighted genes involved in neuroinflammation, G-protein coupled receptor signaling, and cell adhesion.

## Materials and methods

### Animals

All animal procedures were approved by the Institutional Animal Care and Use Committee at Purdue University (Protocol #1912001989). Animals were housed in AAALAC accredited facilities at Purdue University Fort Wayne. Adult male and female Long-Evans rats were purchased from Charles River Laboratories (Wilmington, MA) and were approximately 9 months of age at the time of breeding. All females had previously had successful litters and were not yet in reproductive senescence, which begins between 9–10 months of age [31]. Male and female pairs were co-housed overnight until a sperm plug was observed, designated as G0. At this time, females were removed and single-housed. Dams were assigned to one of three treatment groups: Ad-libitum control (AD), Pair-fed control (PF), and Ethanol-exposed (ET). Ad-libitum control dams were given *ad libitum* access to standard rat chow (Lab Diet, #5001: 23% protein, 4.5% fat, 6% fiber) and were weighed daily throughout gestation. Pair-fed control rats received a control liquid diet (Dyets, Bethlehem, PA; Weinberg-Keiver Heigh Protein Control Diet 710324) and Ethanol-exposed rats received an experimental liquid diet (Dyets, Bethlehem, PA; Weinberg-Keiver Heigh Protein Experimental Diet 710109) with 36% ethanol-derived calories (6.7% ethanol). PF dams were yoked to an ET dam of similar weight, such that the PF dam received the same amount of control diet, adjusted for body weight. To facilitate

**Table 1. Dam and offspring characteristics.** Body weight data for dams (expressed as a % change from G0) and offspring weights on PD 7, 14, 21, and 35–36. Male offspring weighed significantly more than female offspring on PD 7 and on PD 35–36. There were no effects of condition on offspring weights at any age. * p < 0.05 effect of sex.

| Group | Dam Body Weight (% Change) | PD7 weight (g) | PD14 weight (g) | PD21 weight (g) | PD35-36 weight (g) |
|---|---|---|---|---|---|
| ET | 16.62 | M: 15.00 ± 0.58* | M: 29.67 ± 2.40 | M: 48.00 ± 3.79 | M: 176.67 ± 6.11* |
| | | F: 13.50 ± 0.50 | F: 29.50 ± 0.50 | F: 46.50 ± 0.50 | F: 144.00 ± 4.00 |
| PF | 26.59 | M: 15.67 ± 0.49* | M: 29.50 ± 0.92 | M: 50.00 ± 1.53 | M: 157.67 ± 6.36* |
| | | F: 14.67 ± 0.67 | F: 28.25 ± 1.55 | F: 44.00 ± 1.73 | F: 128.67 ± 6.74 |
| AD | 31.56 | M: 15.0 ± 0.63* | M: 28.40 ± 0.93 | M: 47.00 ± 2.08 | M: 159.33 ± 1.45* |
| | | F: 14.4 ± 0.24 | F: 27.00 ± 0.45 | F: 43.33 ± 0.67 | F: 129.67 ± 1.45 |

consumption of the ethanol liquid diet, ET dams were gradually exposed to the diet (Day 1: 33% experimental diet, 67% control diet, Day 2: 67% experimental diet, 33% control diet, Day 3: 100% experimental diet). Fresh diet was provided daily within 1.5–2 hours of lights off [32] and remained on the cage until the diet was changed the next day. Diet was administered throughout gestation (GD 0–20). On G20, all dams were provided water and lab chow (Lab Diet #5001) through the rest of the study. All dams were provided water *ad libitum*.

On Gestational day (GD) 15, a small tail (~ 1 mL) blood sample was taken from a separate set of ET rats (n = 17) at 8:00 pm (3 hours after the diet was placed on the cages) to determine blood ethanol concentrations (BECs). Samples were also taken from PF rats to control for the stress of the tail blood procedure. BECs were assessed using an Analox AM-1 alcohol analyzer (Analox Instruments, Lunenburg, MA). On GD 20, all dams were switched to regular rodent chow (Lab Diet, #5001) and water. As this was an exploratory study to determine if prenatal alcohol exposure resulted in long-term gene expression changes, offspring were derived from a single litter for each treatment group. Body weights were collected from all offspring on postnatal day (PD) 7, 14, 21, and 35–37 and were analyzed using an ANOVA. Offspring were weaned into same-sex groups of 2–3 on PD 21. Dam and litter characteristics can be found in Table 1.

## Hippocampus tissue collection

Rats were 36–38 days of age at time of sample collection. Male and female rats from each treatment group (n = 2-3/group) were removed from their home cage and rapidly decapitated without anesthesia [33–35]. Tissue collection occurred within 3–4 minutes of removing the rat from the home cage, to control for possible changes due to the stress of handling. Whole brains were removed, and hippocampi were dissected on ice, flash frozen, and stored at -80˚C until RNA isolation.

## RNA isolation

RNA was isolated from frozen hippocampal samples stored at -80˚C using the Monarch® RNA Total Isolation kit (NEB T2010S). Hippocampal tissue from the left hemisphere was used where possible, with right hippocampus used for 4 samples. Samples were homogenized frozen in 1X DNA/RNA Protection Reagent (NEB T2011L), then processed according to kit instructions including on-column DNase digestion. Samples were eluted in 50 μL of RNAse-free water and frozen at -80˚C until submitted for sequencing.

## RNA-sequencing

Concentration and purity of RNA was determined using a Nanodrop One C (ThermoFisher) prior to sequencing. Library generation (Illumina TruSeq Stranded mRNA HT Library) and

sequencing (NextSeq 75 –High Output) were performed by the Center for Genomics and Bio-informatics at Indiana University. A total of 17 RNA samples were submitted for sequencing representing the three treatment groups: 5 ET samples (2 females and 3 males), 6 PF samples (3 Females and 3 Males), 6 AD samples (3 Females and 3 Males).

## Quality control and genome mapping

The sequenced data consisted of a total of 51 paired-end sequence reads with three technical replicates for each of the 17 RNA samples. The raw read data was saved in FASTQ-format files where each sequence has approximately 22 million reads. Those reads were pre-processed using Fastp (version 0.23.2); an all-in-one multithreaded high-performance Quality control tool [36]. We employed the default parameters in Fastp where all sequence adapters were auto detected and trimmed. The reads were scanned with the sliding window of 4 and low-quality bases were filtered out if the quality per base drops below 15, where upto 40% of the bases are allowed to be unqualified. The minimum length to detect polyG in the read tail is 10. Furthermore, reads shorter than 15 or those with the number of N base is greater than 5 were discarded.

The resulted filtered reads (approximately 93%) were mapped against the Rattus norvegicus (mRatBN7.2) genome assembly. The read mapping process was performed using Bowtie2 (Version 2.4.2) on the Galaxy platform (https://usegalaxy.org/; version 2.0.1) using the default options [37]. Next, the FeatureCounts tool (part of the Subread package version 2.0.1) was used to quantify the number of reads mapping to the exons of each gene [38]. For a total number of 34322 genes, fragments (or templates) were counted instead of reads. The minimum mapping quality per read was 10 where reads were allowed to map to multiple features. The table resulted from the FeatureCounts tool followed the simplified annotation format (SAF) which contains the counted number of reads (fragments) mapped to each gene (Entrez gene identifiers).

## Differential gene expression analysis

The primary objective of these experiments was to detect transcripts showing differential expressions across various conditions. Among the statistical approaches that have been designed to solve this problem, the DESeq method has been proven to be efficient in finding genes that are differentially expressed between two conditions [39]. DESeq assumes the data follows a negative binomial distribution which best fits the read counts across biological replicates [40].

In this research, we utilized the rnaseqde() function within the *Bioinformatics Toolbox* in *Matlab* (R2022a). As it implements the DESeq method, the rnaseqde() function estimates the biological variance between two conditions using at least two replicates for each. The count data was averaged among the three technical replicates for each sample. The averaged reads were then stored as a table where the rows were labeled with gene identifiers and the columns were labeled with the samples. Prior to performing the hypothesis test, the function applies a normalization step (median-of-ratios) to account for differences in sequencing depth sequencing depth and library composition between samples.

The output of rnaseqde() function is a summary table that includes (for each gene) the mean normalized counts averaged over all samples from both conditions, the fold change in log2, the P-value from the hypothesis test, and the adjusted p-value calculated using a False discovery rate (FDR) method. By default, the function uses the linear step-up procedure introduced by Benjamini and Hochberg. In this study, genes that satisfied FDR adjusted P-value less than 0.01 and an absolute fold change above 1.5 were considered differentially expressed.

## Modifier gene characterization

Information about candidate genes and their human orthologues was gathered from a number of databases including Ratmine, OMIM, and NCBI, then verified through primary sources. Gene ontology categories were derived using the Database for Annotation, Visualization and Integrated Discovery (DAVID) [41, 42]. Gene names and annotation symbols from RNA-seq were converted where possible to Entrez GeneIDs for analysis. This was done using DAVID for most genes, and the rest were identified individually through NCBI Gene (https://www.ncbi.nlm.nih.gov/gene/). Three genes (*LOC100362040*, *LOC100363531*, and *LOC100366054*) did not have current Entrez GeneIDs and were therefore excluded from analysis. Genes that were up (9) or down-regulated (341) in ET rats as compared to either PF or AD rats were included in this analysis. Results from default analyses are included in S12-S25 Tables in S2 File. Categories were considered significant if p < 0.05 after Benjamini multiple testing correction. Trending categories (p < 0.10 after Benjamini multiple testing correction) were included as well.

# Results

## Model characteristics

Using this model, dams consume on average 89.0 (± 8.24) mL (mean ± SD) of the ethanol diet. The ethanol dam used in the current study consumed an average of 88.0 mL of ethanol diet (range = 63–100 mL of ethanol diet), for an average of 5.9 mL of 95% ethanol consumed each night. The average (± SEM) BEC from this group of dams was 44.03 ± 7.21 (Range = 14.90–110.70). Using a similar exposure paradigm, Bodnar et al. (2022) reported BECs of 122.2 ± 26 mg/dL when blood samples were taken at 0700h [43]. Thus, the alcohol exposure here represents consumption in the low-moderate range.

Overall, there were few differences in body weights, with the exception that males weighed significantly more than females on PD 7 [F(1,24) = 4.60, p < 0.05]. No effects of sex or condition were evident at PD 14 or PD 21 (Table 1). There were no significant effects of condition on body weights at the time of tissue collection, although males weighed significantly more than females [F(1,17) = 53.12, p < 0.0001]. Overall, these data were consistent with expectations, with males weighing more than females and no effect of prenatal ethanol exposure.

## RNA-sequencing differentiates ethanol-fed from pair-fed rats

To determine the impact of moderate prenatal ethanol exposure on gene expression in adolescent rats, RNA was isolated from the hippocampus of rats aged 36–38 days. Three dietary groups were compared: the offspring of females were either fed a normal chow diet ad libitum (AD), a liquid diet containing ethanol (ET), or a comparable liquid diet without ethanol (PF). Three males and three females were sampled from each cohort, with the exception of the alcohol-fed litter where only two females were available. Differential gene expression analysis was used to compare all pairwise combinations (adjusted p-value <0.01, Log2Fold-change>1.5), although particular attention was paid to ethanol compared to pair-fed samples since environmental variables related to diet were controlled, with the primary difference being presence or absence of ethanol in the maternal diet (S1-S6 Tables in S2 File). Analysis identified 407 genes that were significantly downregulated upon prenatal exposure to ethanol, while 16 genes were upregulated as compared to PF rats (S1 Table in S2 File).

To determine whether biological sex has a significant impact on the expression of genes that respond to ethanol exposure, we compared expression profiles for males and females under each treatment condition as well as combined. In ET rats, 3 genes were significantly

upregulated in males compared to females and 20 genes were upregulated in females compared to males (S5 Table in S2 File). In PF rats, 6 genes were significantly upregulated in males compared to females and 47 genes were significantly upregulated in females compared to males (S7 Table in S2 File). When these genes were collectively compared to those significantly altered in ET rats compared to PF rats, only one gene (*RT1-N2*) was found be affected by ethanol exposure (downregulated in ET rats) as well as differentially expressed between males and females (upregulated in females compared to males, only after exposure to ethanol). These results are supported by principle components analysis (PCA), which show that sex is not a primary factor segregating the samples (S1 Fig in S1 File). Considering this, we elected to continue with our analysis by pooling male and female samples under the same environmental conditions as conducted in previous studies [30].

PCA did show clustering of samples from the same environmental conditions (Fig 1A). The most distantly separated clusters do appear to be ET and PF groups, with AD samples appearing to bridge the gap between these two. This is consistent with previously published data, which has demonstrated differences in gene expression between PF and AD controls [30, 44]. This clustering lends further support to the primary focus on PF and ET conditions, since these are not only the most directly comparable with regards to environmental conditions, but also appear to be the most differentiated in terms of gene expression.

Additionally, PCA highlighted three potential outlier samples, one from each treatment group. These three samples were separated from the rest of the samples along the first principle component (Fig 1A). To ensure these outlier points did not have undue influence on the results, we removed them from all further analysis. 14 total samples (5 AD, 4 ET, and 5 PF) were compared in the final analysis, in which 393 genes were downregulated upon prenatal exposure to ethanol and 22 genes were upregulated. New PCA demonstrated that environmental condition was still a primary determinant of clustering even after removal of these samples (Fig 1B). While segregation does not occur distinctly along either PC1 or PC2 individually, the groups are visibly clustered in distinct locations on the plot. Furthermore, 96% of these genes are shared with the original list, indicating that removing the outlier samples does not substantially change the overall results (S7-S9 Tables in S2 File).

ET and PF samples were also compared between male samples and female samples independently to confirm the lack of impact of biological sex on results (S10 and S11 Tables in S2 File). For males, this included 4 total samples (2 PF and 2 ET) in which 270 genes were downregulated upon prenatal exposure to ethanol and 25 genes were upregulated (S10 Table in S2 File). For females, this included 5 total samples (3 PF and 2 ET) in which 263 genes were downregulated and 25 genes were upregulated (S11 Table in S2 File). When these lists were compared with genes altered when male and female samples were compared, we found that there was a high degree of overlap (S2 Fig in S1 File). 221 genes were shared from all three comparisons in downregulated genes (56% of compiled comparison, 82% of male comparison, and 84% of female comparison). Very few downregulated genes are unique to male (11/270) or female (9/263) comparisons. 7 genes (*TNNT1*, *SHOX2*, *WNT9B*, *IFIT1*, *SLC39A12*, *NQO2*, *TCAM1*) were shared from all three comparisons in upregulated genes (32% of compiled comparison, 28% of male comparison, and 28% of female comparison). Similarly, very few upregulated genes are unique to male (3/25) or female (1/25) comparisons. The high degree of overlap supports compilation of the samples, as the differentially regulated genes in this study do not appear to be strongly influenced by biological sex. Furthermore, compiling samples and increasing biological replicate number uncovers additional genes (101 downregulated, 6 upregulated), suggesting that it increases the sensitivity and power of the study.

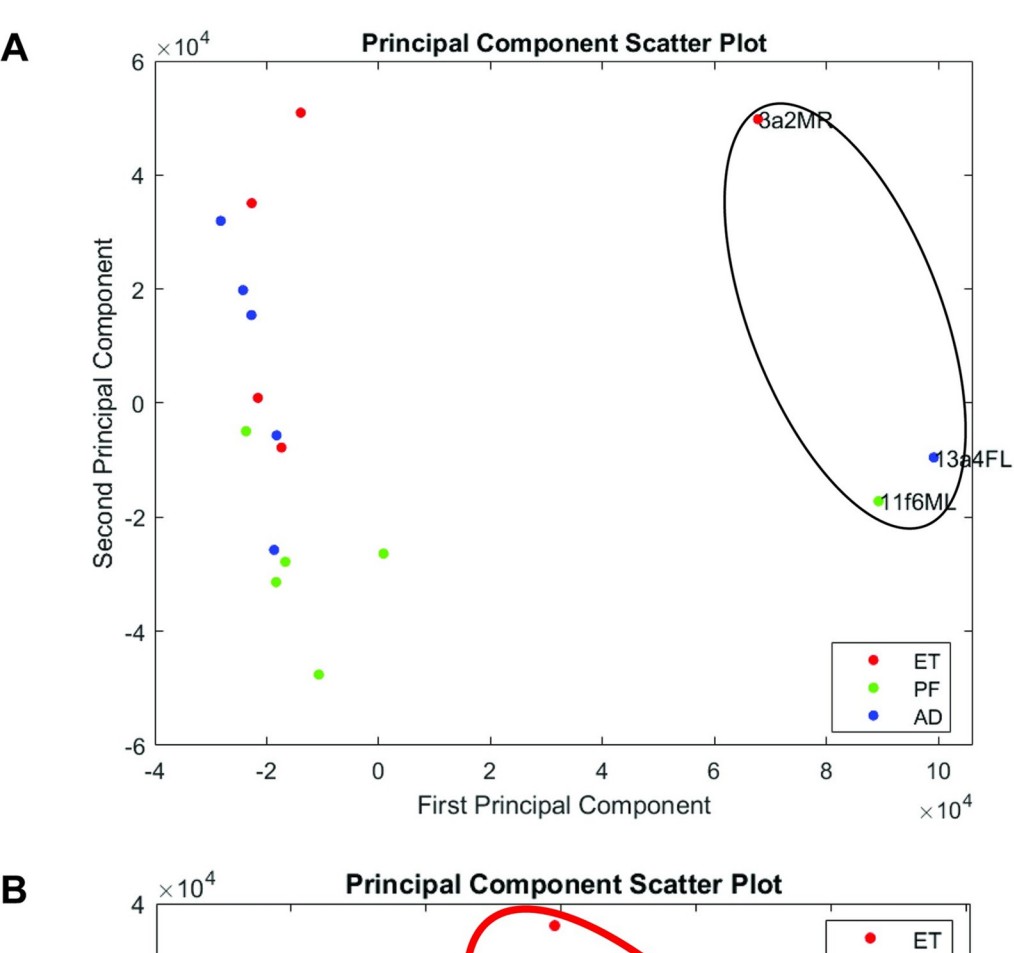

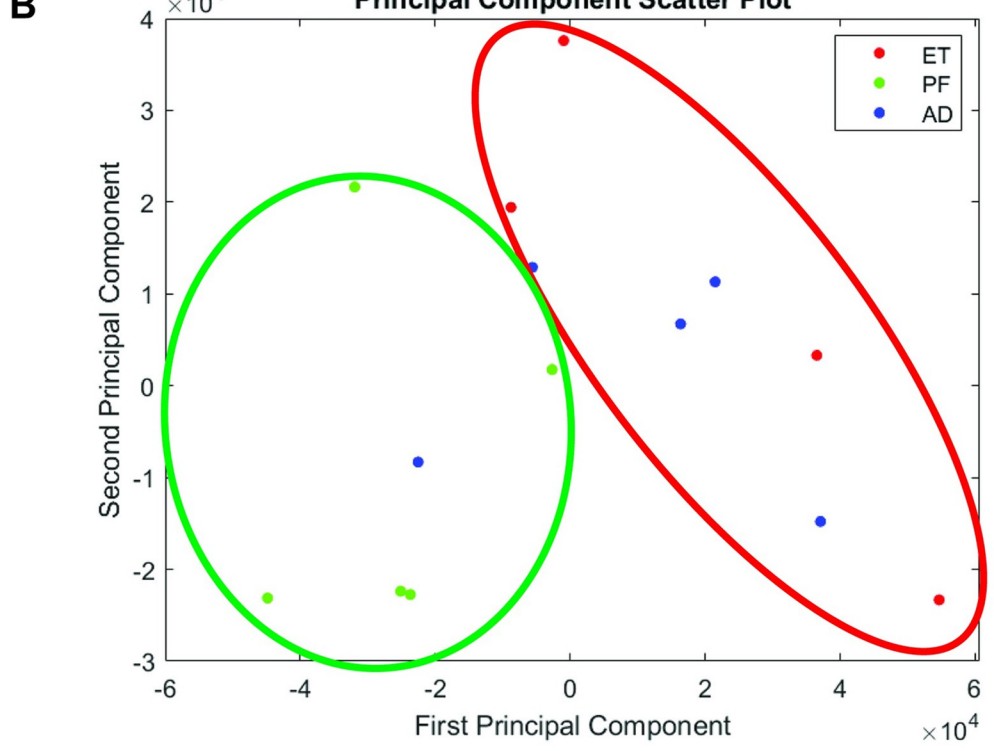

**Figure 1**

**Fig 1. Feeding condition is a primary distinguishing factor in differential gene expression between ethanol-fed and pair-fed mice. A.** Principle components analysis was performed on all 17 RNA-seq samples (first and second principle components). Ethanol exposure samples (ET) are marked in red, pair-fed control samples (PF) are marked in green, and ad libitum-fed control samples (AD) are marked in blue. ET and PF samples primarily segregate along the second principle component, with AD samples segregating between ET and PF. Three outlier samples, one from each group, segregate along the first principle component and are identified by a black circle. **B.** Principle components analysis was performed on all 14 RNA-seq samples after exclusion of the outlier samples marked above (first and second principle components). Ethanol exposure samples (ET) are marked in red (circled in red), pair-fed control samples (PF) are marked in green (circled in green), and ad libitum-fed control samples (AD) are marked in blue. ET and PF samples still segregate, now along both the first and second principle components, with AD samples segregating between ET and PF.

## Genes involved in immunity and cell adhesion are altered after prenatal alcohol exposure

Differential gene expression analysis of the 14 samples remaining after filtering revealed 393 genes that were significantly downregulated upon prenatal exposure to ethanol compared to PF controls, while 22 genes were upregulated (Fig 2A, S7 Table in S2 File). Compared to the AD controls, 410 genes were significantly downregulated upon prenatal exposure to ethanol, while 79 genes were upregulated (Fig 2B, S8 Table in S2 File). Of these genes, 347 are downregulated in comparison to both PF and AD controls, while 16 are upregulated in comparison to both (S7 and S8 Tables in S2 File). The overlap in effect is emphasized in a heat map for the top 10 up and down-regulated genes (ET vs. PF) with annotations as determined by lowest p-value (Fig 3). All ten most downregulated genes share a significant effect for ET compared to PF or AD. Seven of the ten most upregulated genes share a significant effect for ET compared to PF or AD, while three (*ZFP40*, *ZFP458*, and *IFIT1*) are only significantly altered in ET as compared to PF samples. Of these three, *ZFP458* and *IFIT1* both trend toward upregulation in ET as compared to AD samples. Taken together, these analyses suggest that the impact of ethanol feeding is consistent whether comparing to either PF or AD controls.

Of the top ten most downregulated genes with functional annotation, seven belong to the protocadherin gamma subfamily A group of genes (Fig 3). The *PCDHGA* family of genes are encoded in a cluster and function in neuronal cell adhesion, particularly in the CNS, where they may play a role in the regulation of the blood-brain barrier [45]. This is complemented by other cell adhesion genes that are upregulated in response to ethanol exposure, such as *TCAM1*. The identification of cell adhesion in particular serves to validate our study, as previous work has demonstrated that PAE can alter cell-cell interactions through changes in cadherin expression during development [46]. Changes in expression of these genes suggest that prenatal ethanol exposure may lead to changes in neuronal connectivity that persist well past fetal development into adolescence.

Another interesting gene found to be upregulated in ET rats as compared to PF rats is *IFIT1*, a conserved antiviral gene activated upon interferon signaling and commonly found to be upregulated in response to viral infection (Fig 3, S7 Table in S2 File) [47]. This gene also appears to be upregulated in ET rats as compared to AD, although this change is not significant (Fig 3, S8 Table in S2 File). It is well known that alcohol exposure alters the neuroimmune response [48]. Thus, it is not surprising that our analysis identified several genes associated with immunity to be altered following PAE (S7 Table in S2 File).

## Cell adhesion and toxin metabolismare all downregulated after prenatal alcohol exposure

To determine if there are any pathways or processes that are significantly altered in ET rats as compared to both PF and AD control rats when taking into account the entire gene list and

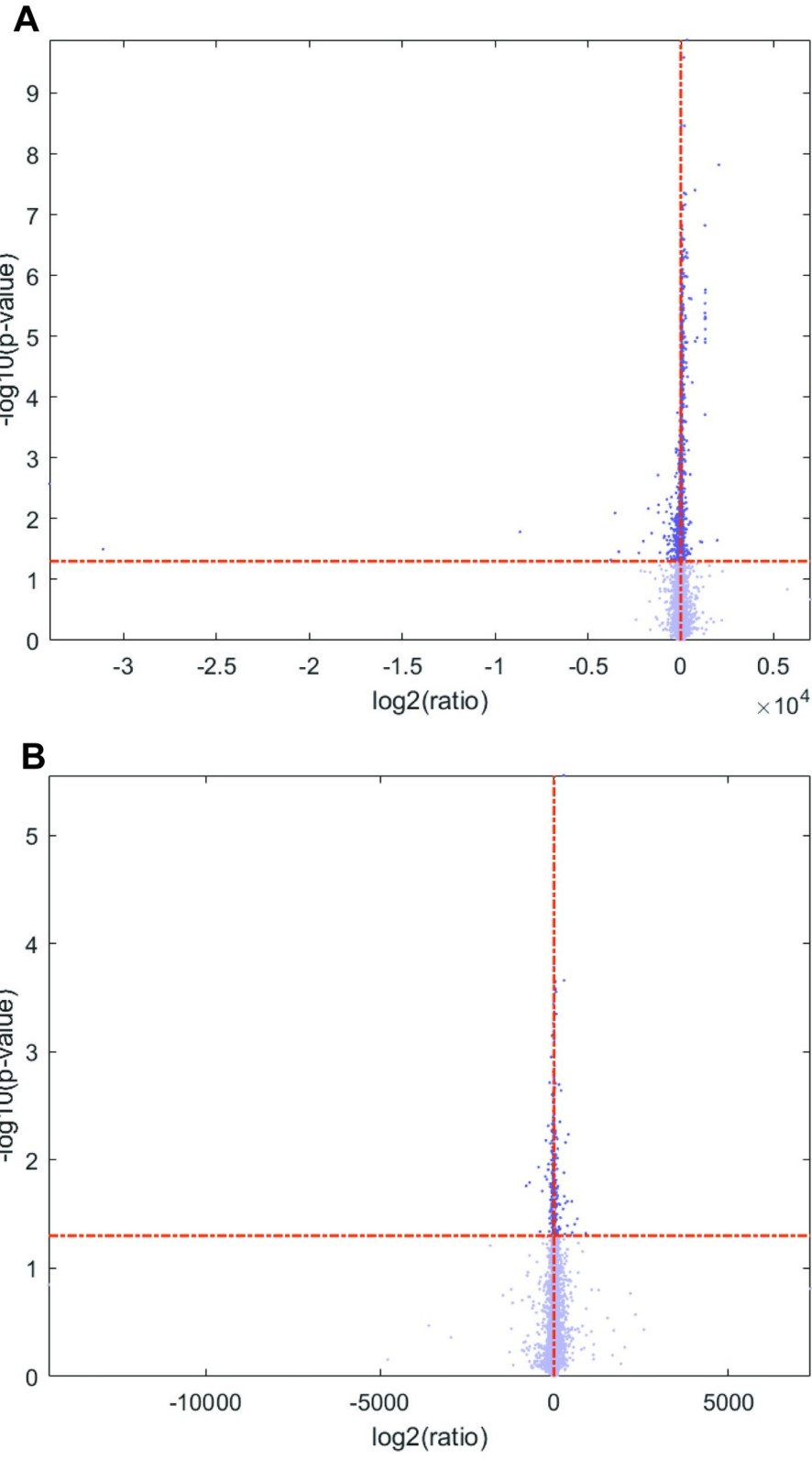

**Figure 2**

**Fig 2. Ethanol exposure significantly alters expression of many genes as compared to pair-fed and ad libitum fed controls. A.** A volcano plot was generated from data in S7 Table in S2 File comparing ET rats to control PF rats. Log2 of the ratio of PF/ET is reported along the X-axis, and the log10 of the adjusted p-value for each change is reported along the Y-axis. Cut-offs for significance were set at an absolute value for Log2FC>1.5 and adjusted p-value of <0.01. These cut-offs are indicated as dotted lines on the plot. In total, 393 genes were significantly downregulated and 22 genes were significantly upregulated in ET rats compared to PF controls. **B.** A volcano plot was generated from data in S8 Table in S2 File comparing ET rats to control AD rats. Log2 of the ratio of AD/ET is reported along the X-axis, and the log10 of the adjusted p-value for each change is reported along the Y-axis. Cut-offs for significance were set at an absolute value for Log2FC>1.5 and adjusted p-value of <0.01. These cut-offs are indicated as dotted lines on the plot. In total, 347 genes were significantly downregulated and 16 genes were significantly upregulated in ET rats compared to AD controls.

not only the top candidates, we performed gene ontology analysis using the Database for Annotation, Visualization and Integrated Discovery (DAVID) [41, 42]. Genes that were upregulated (9) or downregulated (344) in ET rats as compared to both PF and AD controls were included in the analysis. Gene names were converted to Entrez gene IDs for analysis. *LOC100362040*, *LOC100366054*, and *LOC100363531* do not have current Entrez gene IDs and therefore were excluded from this analysis for a total of 341 downregulated genes. No significant categories were identified in any analysis for genes upregulated in ET compared to PF rats (S12 Table in S2 File), likely due to the small number of significant genes identified.

We did observe, however a number of categories across analyses that were enriched in genes downregulated in ET versus both PF and AD control rats (Table 2, S13-S25 Tables in S2 File). We pooled our results across multiple analyses of biological process, molecular function,

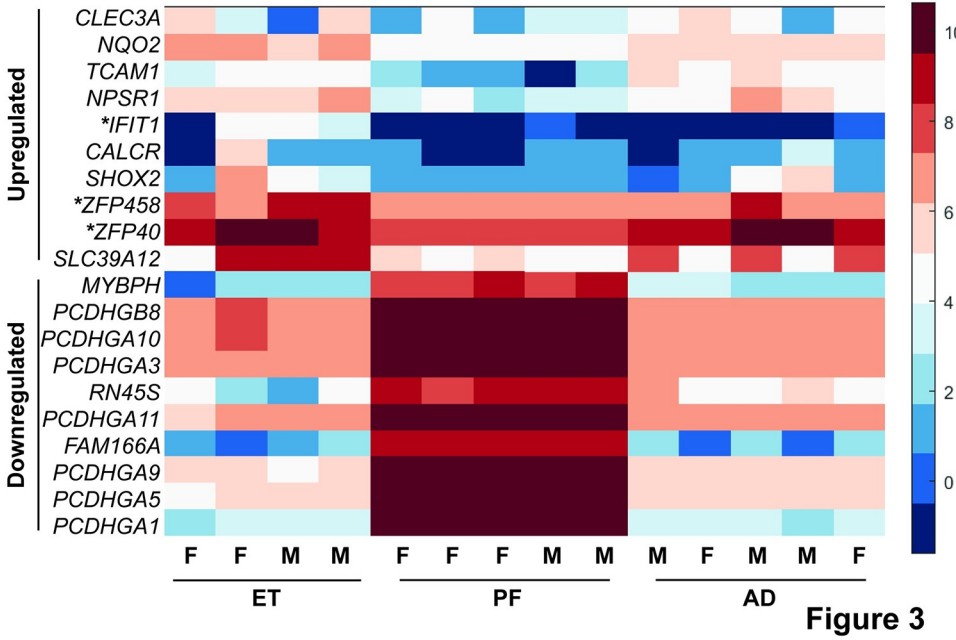

**Fig 3. Top candidate genes differentially expressed in ET rats as compared to PF rats.** A heat map was generated for the top 10 genes with annotations that were differentially expressed in ET rats compared to PF rats as determined by lowest p-value. Unannotated genes are defined as any with a location/annotation symbol only, and without an alternate gene name/function assigned. Individual samples for each group are shown as columns along the X-axis, with the expression of each gene in rows along the Y-axis. Male samples are marked as "M" with female samples marked as "F." All ten downregulated genes are significantly changed in ET as compared to both PF and AD samples. Seven out of ten upregulated genes are significantly changed as compared to both PF and AD samples. Genes that are significantly altered in ET only as compared to PF samples are marked with an *. Blue indicates low relative expression, while red indicates high relative expression (see key, log2(normalized read counts)).

**Table 2. Processes enriched by GO ontology.** General processes highlighted by significant GO categories enriched in genes downregulated in ET rats compared to both control groups (S12-S25 Tables in S2 File). # of List Genes indicates the number of genes from the candidate list that are found in the category. # of Category genes indicates the total number of genes in that category for the genome. Enrichment scores indicate the degree to which the category is represented in the list as compared to the entire genome. P-values reflect the Benjamini corrected values.

| Process of interest | GO Categories | # List genes | # Category genes | Genes | Enrichment | P-value |
|---|---|---|---|---|---|---|
| **Cadherins and Cell Adhesion** | IPR013164: Cadherin, N-terminal | 12 | 54 | *PCDHGB8, PCDHGA3, PCDHGA9, PCDHGA7, PCDHGA11, PCDHGA10, PCDHGA1, PCDHGA12, PCDHGA5, PCDHGA2, PCDHGA8, PCDHGB7* | 47.03 | 1.57E-13 |
| | SM00112: CA | 13 | 103 | *PCDHGA7, PCDHGB8, PCDHGA3, PCDHGA9, PCDHGA11, PCDHGA10, PCDHGA1, CDH23, PCDHGA12, PCDHGA5, PCDHGA8, PCDHGA2, PCDHGB7* | 28.44 | 2.97E-13 |
| | IPR020894: Cadherin conserved site | 13 | 98 | *PCDHGA7, PCDHGB8, PCDHGA3, PCDHGA9, PCDHGA11, PCDHGA10, PCDHGA1, CDH23, PCDHGA12, PCDHGA5, PCDHGA8, PCDHGA2, PCDHGB7* | 28.07 | 2.34E-12 |
| | IPR002126: Cadherin | 13 | 105 | *PCDHGA7, PCDHGB8, PCDHGA3, PCDHGA9, PCDHGA11, PCDHGA10, PCDHGA1, CDH23, PCDHGA12, PCDHGA5, PCDHGA8, PCDHGA2, PCDHGB7* | 26.20 | 3.46E-12 |
| | IPR015919: Cadherin-like | 13 | 107 | *PCDHGA7, PCDHGB8, PCDHGA3, PCDHGA9, PCDHGA11, PCDHGA10, PCDHGA1, CDH23, PCDHGA12, PCDHGA5, PCDHGA8, PCDHGA2, PCDHGB7* | 25.71 | 3.46E-12 |
| | DOMAIN: Cadherin | 13 | 107 | *PCDHGA7, PCDHGB8, PCDHGA3, PCDHGA9, PCDHGA11, PCDHGA10, PCDHGA1, CDH23, PCDHGA12, PCDHGA5, PCDHGA8, PCDHGA2, PCDHGB7* | 27.58 | 6.93E-12 |
| | KW-0130: Cell adhesion | 17 | 304 | *PCDHGA7, MYBPH, GLYCAM1, PCDHGB8, PCDHGA3, PCDHGA9, PCDHGA11, PCDHGA10, PCDHGA1, IGFALS, CDH23, PCDHGA12, ICAM2, PCDHGA5, PCDHGA8, PCDHGA2, PCDHGB7* | 9.69 | 9.42E-11 |
| | GO:0007156: homophilic cell adhesion via plasma membrane adhesion molecules | 13 | 155 | *PCDHGA7, PCDHGB8, PCDHGA3, PCDHGA9, PCDHGA11, PCDHGA10, PCDHGA1, CDH23, PCDHGA12, PCDHGA5, PCDHGA8, PCDHGA2, PCDHGB7* | 18.55 | 1.86E-09 |
| **Glucuronidation** | rno00040: Pentose and glucuronate interconversions | 4 | 35 | *AKR1B7, UGT1A6, UGT1A1, UGT1A8* | 34.95 | 1.15E-02 |
| | GO:0052696: flavonoid glucuronidation | 3 | 8 | *UGT1A6, UGT1A1, UGT1A8* | 82.92 | 9.71E-02 |
| | GO:0052697: xenobiotic glucuronidation | 3 | 9 | *UGT1A6, UGT1A1, UGT1A8* | 73.71 | 9.71E-02 |
| **Toxin Response** | rno00982: Drug metabolism—cytochrome P450 | 4 | 73 | *UGT1A6, UGT1A1, GSTM3I, UGT1A8* | 16.76 | 2.26E-02 |
| | rno00980: Metabolism of xenobiotics by cytochrome P450 | 4 | 76 | *UGT1A6, UGT1A1, GSTM3I, UGT1A8* | 16.10 | 2.26E-02 |
| | rno05204: Chemical carcinogenesis—DNA adducts | 4 | 76 | *UGT1A6, UGT1A1, GSTM3I, UGT1A8* | 16.10 | 2.26E-02 |
| | rno00983: Drug metabolism—other enzymes | 4 | 97 | *UGT1A6, UGT1A1, GSTM3I, UGT1A8* | 12.61 | 3.75E-02 |
| | rno05207: Chemical carcinogenesis—receptor activation | 5 | 228 | *NR1I3, UGT1A6, UGT1A1, GSTM3I, UGT1A8* | 6.71 | 4.47E-02 |
| **Calcium ion binding** | GO:0005509: calcium ion binding | 15 | 704 | *SPATA21, PCDHGA7, PD2I2, PCDHGB8, PCDHGA3, PCDHGA9, PCDHGA11, PCDHGA10, PCDHGA1, CDH23, PCDHGA12, PCDHGA5, PCDHGA8, PCDHGA2, PCDHGB7* | 5.05 | 1.40E-04 |

*(Continued)*

**Table 2.** (Continued)

| Process of interest | GO Categories | # List genes | # Category genes | Genes | Enrichment | P-value |
|---|---|---|---|---|---|---|
| **Misc Metabolic Processes** | rno00970: Aminoacyl-tRNA biosynthesis | 4 | 67 | *TRNN, TRNF, TRNC, TRNA* | 18.26 | 2.26E-02 |
| | rno00053: Ascorbate and aldarate metabolism | 3 | 30 | *UGT1A6, UGT1A1, UGT1A8* | 30.58 | 3.75E-02 |
| | rno00860: Porphyrin metabolism | 3 | 43 | *UGT1A6, UGT1A1, UGT1A8* | 21.34 | 5.90E-02 |
| | rno01240: Biosynthesis of cofactors | 4 | 154 | *301115, UGT1A6, UGT1A1, UGT1A8* | 7.94 | 8.12E-02 |
| **Membrane Proteins** | KW-0812: Transmembrane | 41 | 5334 | *PCDHGA7, TMEM262, TCTE1, CLCN1, FAM151A, LRRN4, LAIR1, SLC3A1, ACP4, PCDHGA12, PCDHGA5, UGT1A8, TMEM150B, ERGIC2, CLEC2DL1, KCNK15, SLC5A2, DEAR, UGT1A1, PD2I2, SPATA9, SLC24A5, PCDHGB8, PCDHGA3, PCDHGA9, CHRNB1, PCDHGA11, PCDHGA10, SERINC4, NOX1, PCDHGA1, UGT1A6, CDH23, ICAM2, TMEM54, LOC687508, PCDHGA2, PCDHGA8, RGD1563263, LMLN2, PCDHGB7* | 1.44 | 3.68E-02 |
| | KW-0325: Glycoprotein | 16 | 2685 | *FAM20A, CLEC2DL1, SLC5A2, UGT1A1, TGM4, PD2I2, GLYCAM1, LRRN4, LAIR1, SLC3A1, CHRNB1, NOX1, IGFALS, UGT1A6, CDH23, UGT1A8* | 1.53 | 4.23E-01 |
| **Repeat Domains** | KW-0677: Repeat | 26 | 3058 | *PCDHGA7, MYBPH, CLCN1, LRRN4, IGFALS, ESRP2, PCDHGA12, PCDHGA5, LOC290428, RCOR2, WDR38, DPH7, ASB14, PCDHGB8, PCDHGA3, PCDHGA9, LRRC39, PCDHGA11, PCDHGA10, MORN5, PCDHGA1, CDH23, SNRPN, PCDHGA2, PCDHGA8, PCDHGB7* | 1.60 | 7.89E-02 |

pathway enrichment, and protein domains for further investigation (Table 2). Of particular interest was enrichment for cell adhesion and cadherin genes (GO:0007156; KW-0130, IPR013164, IPR020894, IPR002126, IPR015919, Cadherin, SM00112:CA, KW-0812, KW-0325), glucuronidation (GO:0052696, GO:0052697, rno00040, KW-0325), and toxin metabolism (rno00982, rno00980, rno05204, rno00983, rno05207) (Table 2, S13-S25 Tables in S2 File). In all of these, multiple categories were identified through differing analyses, strengthening their association in this study. The further validation of cell adhesion as responding to PAE once more serves to validate our study [46]. Our work supports these conclusions and furthermore indicates that they persist long after initial alcohol exposure, continuing into adolescence. Genes in the highlighted GO categories that achieved statistical significance were compiled into a comprehensive list of 51genes (Fig 4 and S2 Fig in S1 File). Of particular note are the *PCDHG* cell adhesion genes (12 of the 51 GO highlighted genes), as well as the UDP-glucuronosyltransferases (*UGT*) (3 of the 51 GO highlighted genes) (Fig 4 and S2 Fig in S1 File). We note the similar changed that are found in genes of the same category for each of the highlighted genes, lending greater support to the importance of each highlighted process in the response to ethanol exposure.

Glucuronidation was another interesting process highlighted by several analyses and categories (GO:0052696, GO:0052697, rno00040, KW-0325) in addition to a number of categories that did not retain statistical significance after multiple testing correction (S13-S25 Tables in S2 File). This pathway modifies various molecules with glucuronate, a sugar acid salt derived from glucose and most commonly involved in detoxifying and defense responses [49]. The downregulation of other toxin and drug metabolizing genes and pathways (rno00982, rno00980, rno05204, rno00983, rno05207) in conjunction with glucuronidation suggests that the neurons continue to be affected by early toxin exposure in the form of alcohol.

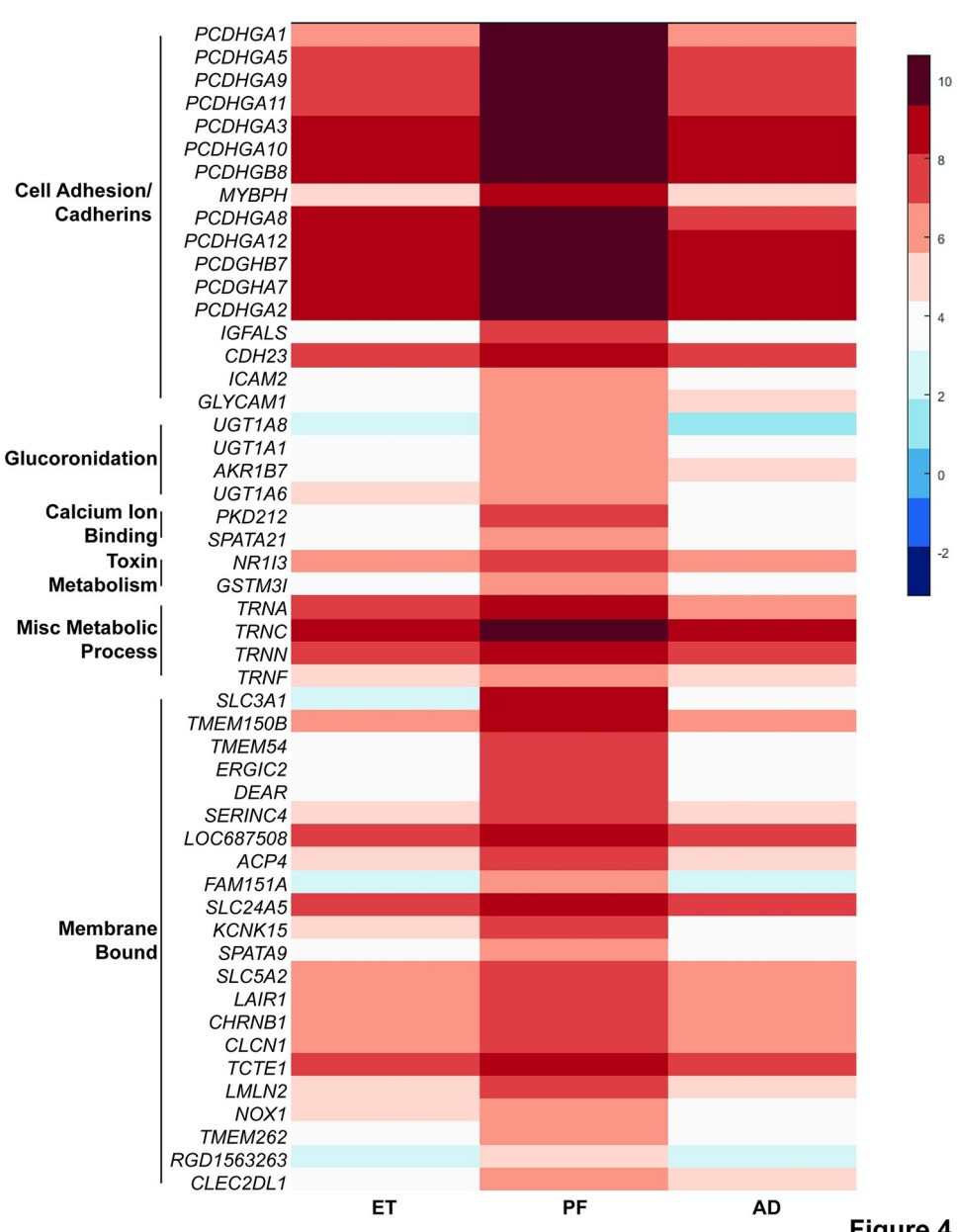

**Figure 4**

**Fig 4. Genes from highlighted GO categories that are differentially expressed in ET rats as compared to PF rats.** A heat map was generated for the 51 genes that were differentially expressed in ET rats compared to both PF and AD rats and were found in the significant GO categories from Table 2. Treatment categories are shown along the X-axis, with the average expression of each gene for that group in rows along the Y-axis. Represented process is labeled on the Y-axis. Blue indicates low relative expression, while red indicates high relative expression (see key, average log2 (normalized read counts) across all individuals in a group).

## Discussion

Here we demonstrated that PAE produced persistent downregulation of gene expression in the hippocampus. Gene ontology analysis was used to determine the categories of genes that were affected by prenatal alcohol treatment. Several of these categories are of interest, with PAE reducing the expression of genes involved in cell adhesion, glucuronidation, toxin

metabolism, and the immune response. PCA analysis supports these findings, demonstrating visual separation of groups on the basis of exposure and diet but not on the basis of sex. This segregation was visually apparent despite the fact that the individual principle components did not fully segregate groups. In addition, few sex differences were evident, with only one gene (*RT1-N2*) upregulated in ET females compared to ET males. *RT1-N2* is predicted to play a role in antigen processing and is orthologous to human major histocompatibility complex class 1. Previous studies evaluating sex differences in gene expression following PAE in the CNS have had mixed results. For example, Gano et al. (2020) found no evidence of sex differences in the olfactory bulb of adolescent rats that were exposed to alcohol on GD11-12 [30]. However, Mishra et al. (2023) reported that over 1,000 genes were differentially expressed in the cortex of male and female adolescent mice prenatally exposed to alcohol [19]. It is possible that sex differences in gene expression vary by brain region. Although most genes were downregulated following PAE, expression of *IFIT1* was upregulated in ET rats. *IFIT1* is part of the interferon-induced proteins with tetratricopeptide repeats (IFIT) family, where it participates in nonspecific antiviral response [47]. It is somewhat surprising that PAE caused a long-lasting increase in the expression of *IFIT1* but supports other studies showing long-term programming of the immune response by alcohol [30, 48].

We observed that PAE resulted in downregulation of several protocadherins (PCDH) from the gamma subfamily. It is not surprising that PAE resulted in alterations in cell adhesion, as prior research has reported that alcohol exposure alters expression of neural cell adhesion molecule (NCAM) in zebrafish (100mM ethanol) [50] and in primary cultures of cortical neurons (400 mg/dL) [51]. In addition, the neural cell adhesion molecule L1 is altered by alcohol [52, 53]. However, we did not directly observe a change in expression of NCAM or L1. Instead, there was a significant downregulation of several protocadherins, another important class of cell adhesion molecules with known roles in cell-cell interactions [54] and previously shown to be altered by developmental alcohol exposure [46]. For example, in humans with FASD, there is an increase in DNA methylation of genes related to protocadherins in buccal samples [23, 55]. In addition, protocadherin 18a was decreased following alcohol exposure (100mM) in a zebrafish model of FASD [56], indicating that altered expression of protocadherins may be a consistent consequence of prenatal alcohol exposure. PCDH are a subtype of cadherins, a superfamily of proteins involved in cell-cell interactions, that are expressed extensively in the central nervous system [54]. PCDH can be divided into several families: α, β, and γ, to name a few [54]. Of these, the γ-protocadherins (PCDH-γ) are of particular interest, due to their role in synaptogenesis [57] and blood-brain barrier formation [45]. Previous studies have found that PAE does not affect the expression of synaptophysin, PSD-95, or hippocampal spine density at PD30 [58, 59] or at PD60 [59]. Thus, it is not likely that PAE results in long-term effects on synapse formation. However, PCDH-γ are expressed in endothelial cells of the blood-brain barrier [45] and in astrocytes [60]. Previous studies have reported that PAE disrupts brain vasculature in the hippocampus [61] and cortex [62]. As such, it is possible that PAE disrupts blood-brain barrier integrity through a decrease in the expression of *PCDH-γ*. Although we did not evaluate gene expression in different cell types, it would be interesting to examine *PCDH-γ* expression in astrocytes, specifically. Astrocytes contribute to the formation of the blood brain barrier, and PCDH-γ are important for neuron-astrocyte interactions [60].

We also noted differential expression of several genes involved in glucuronidation. This pathway modifies various molecules with glucuronate, a sugar acid salt derived from glucose and most commonly involved in detoxifying and defense responses [49]. During glucuronidation, the enzymes involved utilize UDP-glucuronate as a cofactor to add the sugar salt to the target. These molecules then are transported out of the cell, usually to be excreted in urine, bile, or feces. Common targets include toxins such as xenobiotics, drugs, and other

environmental or dietary factors harmful to the organism [49]. The downregulation of other toxin and drug metabolizing genes and pathways in conjunction with glucuronidation suggests that the neurons continue to be affected by early toxin exposure in the form of alcohol.

We observed significant differences in the expression of several genes involved in the immune response. First of all, several members of the *RT1* complex (rat MHC) class 1 family of genes were downregulated in the hippocampus of ET rats compared to PF rats *(RT1-M1-2, RT1-O1, RT1-N2, RT1-T24-4)*. The *RT1* complex is a large cluster of genes, some of which are orthologous to human major histocompatibility complex (MHC). In the rat, the *RT1* complex can be further divided into framework, Class 1, and Class II genes [63, 64]. MHC class 1 molecules and their receptors are expressed in many neuronal subtypes as well as in microglia, where they are thought to play a pivotal role in the refinement of synapses [65]. Thus, the observed downregulation of *RT1* class 1 genes following PAE suggests that important developmental processes, such as synaptic pruning and stripping, may be impaired by PAE.

Another important component of the immune response involves neutrophil transfer across the blood-brain barrier, a process that relies on intercellular cell adhesion molecule (ICAM)-1 and ICAM-2 [66]. ICAM-2 is expressed on endothelial cells [67, 68] and plays a role in T-cell diapedesis through the blood-brain barrier [69]. Furthermore, an analysis of fetal expression of *ICAM-2* revealed colocalization with microglia in the corpus callosum and cerebral vessels of the cortex [70], although this study observed low expression of *ICAM-2* in the adult brain. However, a study by Navratil et al. (1997) reported significant expression of *ICAM-2* in vascular endothelial cells of adult post-mortem tissue [71]. We observed a significant decrease in *ICAM-2* following PAE, which when combined with the observed downregulation of *PCDH-γ* could indicate significant dysfunction of the blood-brain barrier in alcohol-exposed rats. A previous study found that PAE resulted in altered angiogenesis and blood-brain barrier composition in newborn mice [62], whether these changes persist into adolescence remains to be seen. The data presented here suggests that PAE could indeed result in long-term changes to the integrity of the blood-brain barrier.

As with any study, there are limitations to the current data. First, we elected to evaluate gene expression in the hippocampus only, while acknowledging that gene expression, and its changes following PAE are likely to be region dependent. However, changes in gene expression of protocadherins following developmental exposure have been observed in mouse cortex [19], human buccal samples [23, 55], and zebrafish embryo [56], suggesting that this may be a reliable consequence of prenatal alcohol exposure. Another important factor to consider is the timing of alcohol exposure. In the rodent, prenatal alcohol administration results in exposure that is limited to the first and second trimester periods. It is entirely possible that a different outcome would be observed with postnatal exposure during the brain growth spurt (PD 4–9). Future studies should evaluate whether the timing of exposure is an important factor in the current outcome. Finally, there are limitations in the study design, as samples were derived from a single litter per treatment group. In this way, the current data should be viewed as exploratory. Furthermore, we were limited in the number of samples that could be analyzed in the current study. After finding no sex differences, we elected to combine males and females, resulting in n's of 5-6/group (2–3 males and females).We recognize these factors as limitations of the current work and plan to replicate and expand these studies going forward. However, the current findings align well with the existing literature, providing support for the current data. Finally, future studies should validate the expression changes observed in the current study.

Taken together, it is clear that PAE causes enduring changes to the hippocampal genome. Almost 400 genes were downregulated in ET rats compared to PF rats, with GO analysis indicating enrichment of cell adhesion and cadherin genes, genes involved in toxin metabolism,

and genes with a role in the immune response. Of note is the observation that several *PCDH-γ* and *RT1* complex genes were downregulated by alcohol exposure. Given the role of these genes in synapse formation and synaptic refinement, it is possible that PAE impairs critical processes such as synaptic pruning within the hippocampus. The hippocampus plays a major role in synaptic plasticity and memory consolidation, so impaired synaptic pruning within this structure could contribute to cognitive dysfunction in individuals with FASD.

## Supporting information

**S1 File.**
(PDF)

**S2 File.**
(XLSX)

**S1 Data.**
(CSV)

## Acknowledgments

We would like to thank Carolyn Pang and Jayde Bransteter for their contributions to this project.

## Author Contributions

**Conceptualization:** Amal Khalifa, Rebecca Palu, Amy E. Perkins.

**Data curation:** Amal Khalifa, Rebecca Palu, Amy E. Perkins.

**Formal analysis:** Amal Khalifa, Rebecca Palu, Amy E. Perkins.

**Funding acquisition:** Amal Khalifa, Rebecca Palu, Amy E. Perkins.

**Investigation:** Amal Khalifa, Rebecca Palu, Amy E. Perkins, Avery Volz.

**Methodology:** Amal Khalifa, Rebecca Palu, Amy E. Perkins.

**Project administration:** Amal Khalifa, Rebecca Palu, Amy E. Perkins.

**Resources:** Amal Khalifa, Rebecca Palu, Amy E. Perkins.

**Software:** Amal Khalifa.

**Supervision:** Amal Khalifa, Rebecca Palu, Amy E. Perkins.

**Validation:** Amal Khalifa, Rebecca Palu, Amy E. Perkins, Avery Volz.

**Visualization:** Amal Khalifa, Rebecca Palu, Amy E. Perkins.

**Writing – original draft:** Amal Khalifa, Rebecca Palu, Amy E. Perkins.

**Writing – review & editing:** Amal Khalifa, Rebecca Palu, Amy E. Perkins, Avery Volz.

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
