## [Decision Letter · Decision Letter 0]

17 Jul 2023

PONE-D-23-18670Prenatal alcohol exposure alters expression of genes involved in cell adhesion, immune response, and toxin metabolism in adolescent rat hippocampusPLOS ONE

  Dear Dr. Palu,

Thank you for submitting your manuscript to PLOS ONE. After careful consideration, we feel that it has merit but does not fully meet PLOS ONE’s publication criteria as it currently stands. Therefore, we invite you to submit a revised version of the manuscript that addresses the points raised during the review process.

ACADEMIC EDITOR: The current version of this manuscript requires substantial revisions. There is a lack of interpretation and discussion of the current literature in the Introduction. In addition, many papers were not included in the current version of the manuscript. Other specific comments raised by reviewers need to be addressed. 

We look forward to receiving your revised manuscript.

Kind regards,

Jia Luo, Ph.D.

Academic Editor

PLOS ONE

“We would like to thank Carolyn Pang and Jayde Bransteter for their contributions to this project. Funding provided by Purdue University Fort Wayne through a Collaborative Research Grant from Purdue University Fort Wayne to AK, RASP, and AP. Support was also provided to RASP and AP from the Purdue University Fort Wayne Departments of Biology and Psychology.”

“Funding provided by Purdue University Fort Wayne through a Collaborative Research Grant from Purdue University Fort Wayne (https://www.pfw.edu/offices/sponsored-programs/internal-grants-funding/collaborative-research-grant) to AK, RASP, and AP. The funders had no role in study design, data collection and analysis, decision to publish, or preparation of the manuscript.”

Reviewers' comments:

Reviewer's Responses to Questions

**Comments to the Author**

1. Is the manuscript technically sound, and do the data support the conclusions?

Reviewer #1: No

Reviewer #2: Partly

2. Has the statistical analysis been performed appropriately and rigorously? 

Reviewer #1: No

Reviewer #2: Yes

3. Have the authors made all data underlying the findings in their manuscript fully available?

Reviewer #1: Yes

Reviewer #2: Yes

4. Is the manuscript presented in an intelligible fashion and written in standard English?

Reviewer #1: Yes

Reviewer #2: Yes

5. Review Comments to the Author

Reviewer #1: The manuscript titled “Prenatal alcohol exposure alters expression of genes involved in cell adhesion, immune response, and toxin metabolism in adolescent rat hippocampus” by Khalifa and colleagues aimed to characterize the long-term effects of prenatal alcohol exposure on gene expression in rat hippocampus, reporting significant differences in the expression of genes involved in the cell adhesion process and immune response. The current version of the manuscript needs extensive revision before acceptance for publication.

Major comments

• In the introduction section, the authors summarized the current knowledge about prenatal alcohol exposure (PAE) effects, but in the current version, it is not clear the rationale of the present investigation, in particular why they decided to focus their attention on neuroinflammation, G-protein coupled receptor signaling, and cell adhesion. In the last part of the introduction, the authors discussed the data collected to date about the hippocampus and their limitations (sex-variable, age). However, it would be better for validating the importance of the current study to cite previous experimental evidence performed in vitro models (studying the cell-adhesion class) and discuss the importance of alcohol concentration and/or paradigm used, in order to highlight what the present investigation is going to add to the current knowledge.

• In the same section, the authors reported the current findings about PAE effects on gene expression, but for different studies, they did not report the paradigm used (acute vs long exposure), the alcohol concentration used, voluntary drink vs gavage.

• The authors from lines 66 to 69 listed the neurobehavioral outcomes reported induced by PAE. It would be more accurate also to include the references.

• In the Results section titled “RNA-sequencing differentiates ethanol-fed from pair-fed rats”, it is reported the BEC collected during prenatal alcohol exposure, BEC range: 14.90 -110.70; mean: 44.03 ± 7.21. Unfortunately, it is not mentioned the unit used for the measurement both here and in the Material Methods Section. Furthermore, they are talking about moderate PAE, but usually, for a moderate model, the BEC is around 80-90 mg/dl. Please clarify this point.

• As mentioned previously, the PAE paradigm used is described in the Material and Methods section. The current version is not clear to readers, it would be better to add more details such as the age of female rats before starting with the paradigm, and the number of litters used. It seems that for each experimental group, the authors used one litter since it is reported in line 134 “ with the exception of the alcohol-fed litter where only two females were available”. Please clarify this point.

• Also, the readers have to assume that the exposure to alcohol started during the first day of gestation (G1) until G20. Please add more details about the protocol. Any specific reason for choosing this period of time?

• Furthermore, it would be more accurate to report the concentration of Ethanol used in the experimental diet and the average consumption during 1.5-2 hours of exposure- drinking in the dark. Furthermore, it is not clear if the dams were exposed for 1.5 or 2 hours. Clarify this critical point.

• In the same section, authors reported the analysis performed for comparing possible differences in the body weight of pups at P7, showing a significant difference between males and females. Add more details about the statistical test used for those comparisons.

• In the section “Hippocampus tissue collection”, the authors reported that the animals were decapitated without anesthesia. Why did they decide to use this procedure? The stress during the sacrifice could be affected the result. Do the authors consider the possible contamination of blood?

• In the discussion section, the authors compared their findings with the current literature talking about differences related to the brain region examined and sex differences. However, they did not mention the possible differences related to the prenatal alcohol exposure paradigm used (prenatal vs 3rd trimester), the alcohol concentration.

Minor comments

1. The authors across the entire manuscript use the abbreviation FASD for Fetal Alcohol Spectrum Disorders as indicated in line 29, since they are using the plural of disorder, it would be more correct to add the “s” to the abbreviation FASD.

2. Please indicate the acronym for Fetal Alcohol Syndrome (FAS).

3. Several times, the authors did not spell out the acronyms used, for example, it would be more appropriate to define MRI (line 74), DTI (line 76).

4. In the Author Summary section, it is used the term mothers for the rat dams. It would be more appropriate to use the term dams.

5. In line 67, the authors defined the acronym PAE for Prenatal Alcohol Exposure, but they did not use it in the entire manuscript. Please be consistent in using the acronyms. Lines: 81, 90, 100, 106, 109, 111, 114, 119, 122, 129, 249, 268, 274, 278, 280, 282, 308, 311, 313, 315, 320.

6. When they are going to indicate the age of the animals, explain for the first time the meaning of P indicating postnatal day.

7. Table 1: typo for “adhesion”

Reviewer #2: In this manuscript, Amal Khalifa, et al. isolated RNA from the hippocampus of adolescent rats exposed to ethanol during prenatal development and compared gene expression to controls. They found many genes were downregulated and these genes were associated with cell adhesion, toxin metabolism, and immune responses. They concluded from this study that changes in genetic architecture after prenatal exposure to alcohol continue through adolescent development.

There are some concerns.

1. From fetus to adolescence, there are many factors that could affect the changes in gene expression. Even among littermates, gene expression can vary significantly. Therefore, in order to identify the effect of fetal alcohol exposure on adolescent gene expression, a relatively large sample size is required to eliminate the influence of these factors. In this study, three males and three females were sampled for each group, with the exception of the alcohol-fed group that only two females were sampled. When analysis, three outlier samples were removed. The results from a small sample size study are likely affected by many factors, resulting in false positive results.

2. For the most differentially expressed genes, the expression and function of these genes of interest need to be further validated.

3. As for sex differences, in Table S5, the manuscript compared ET Males with ET Females. In Table S6, compared PF Males with PF Females. As readers and reviewers, we also hope to know the comparison between ET Males and PT Males and the comparison between ET Females and PT Females. I think the authors should perform the comparisons and include them into the manuscript, even though the principle components analysis (PCA) shows that sex is not a primary factor.

4. On page 25, in the first line, in Figure 2B, the volcano plot was generated from data comparing ET rats to AD rats, not PF rats. This might be a typo.

6. PLOS authors have the option to publish the peer review history of their article (what does this mean?). If published, this will include your full peer review and any attached files.

Reviewer #1: No

Reviewer #2: No

---

## [Author Response · Author response to Decision Letter 0]

9 Aug 2023

PONE-D-23-18670

Response to Reviewers

 We thank the reviewers for their insightful comments and have provided a point-by-point response below. 

Comments to the Author

1. Is the manuscript technically sound, and do the data support the conclusions?

Reviewer #1: No

Reviewer #2: Partly

2. Has the statistical analysis been performed appropriately and rigorously?

Reviewer #1: No

Reviewer #2: Yes

3. Have the authors made all data underlying the findings in their manuscript fully available?

Reviewer #1: Yes

Reviewer #2: Yes

4. Is the manuscript presented in an intelligible fashion and written in standard English?

Reviewer #1: Yes

Reviewer #2: Yes

5. Review Comments to the Author

Reviewer #1: The manuscript titled “Prenatal alcohol exposure alters expression of genes involved in cell adhesion, immune response, and toxin metabolism in adolescent rat hippocampus” by Khalifa and colleagues aimed to characterize the long-term effects of prenatal alcohol exposure on gene expression in rat hippocampus, reporting significant differences in the expression of genes involved in the cell adhesion process and immune response. The current version of the manuscript needs extensive revision before acceptance for publication.

Major comments

• In the introduction section, the authors summarized the current knowledge about prenatal alcohol exposure (PAE) effects, but in the current version, it is not clear the rationale of the present investigation, in particular why they decided to focus their attention on neuroinflammation, G-protein coupled receptor signaling, and cell adhesion. In the last part of the introduction, the authors discussed the data collected to date about the hippocampus and their limitations (sex-variable, age). However, it would be better for validating the importance of the current study to cite previous experimental evidence performed in vitro models (studying the cell-adhesion class) and discuss the importance of alcohol concentration and/or paradigm used, in order to highlight what the present investigation is going to add to the current knowledge.

- We apologize for the confusion regarding the analyses, but we did not choose to focus on any specific gene targets for the analyses in this manuscript. We conducted an unbiased screen of changes in gene expression and subsequently conducted a gene ontology analysis that revealed significant changes in gene expression in cell adhesion, toxin metabolism, and immune processes. We discuss the relevance of these changes in the discussion. 

- We have added some references to the introduction and discussion focusing on protocadherins. Importantly, previous studies in humans have shown increased DNA methylation of these genes, lending support to the current work. 

- We respectfully disagree that a discussion of in vitro models in the introduction is a better justification for the importance of the current study. The current study uses an in vivo model of prenatal alcohol exposure and the literature provided in the introduction highlights findings from studies with similar methodologies. However, we do feel that including a better discussion of the findings of studies measuring cell adhesion is warranted. To this end, we have elected to expand the discussion (Lines 330-355 the revised version) to include these important references. We thank the reviewer for this insightful suggestion and think that these changes greatly improve the discussion. 

• In the same section, the authors reported the current findings about PAE effects on gene expression, but for different studies, they did not report the paradigm used (acute vs long exposure), the alcohol concentration used, voluntary drink vs gavage.

- We have added some of these details to the current manuscript to address these concerns (Lines 101-120 of revised manuscript). 

• The authors from lines 66 to 69 listed the neurobehavioral outcomes reported induced by PAE. It would be more accurate also to include the references.

- We apologize for this oversight and have added the appropriate reference (Mattson et al., 2019).

• In the Results section titled “RNA-sequencing differentiates ethanol-fed from pair-fed rats”, it is reported the BEC collected during prenatal alcohol exposure, BEC range: 14.90 -110.70; mean: 44.03 ± 7.21. Unfortunately, it is not mentioned the unit used for the measurement both here and in the Material Methods Section. Furthermore, they are talking about moderate PAE, but usually, for a moderate model, the BEC is around 80-90 mg/dl. Please clarify this point.

- We obtained blood samples from a separate set of dams given the ethanol diet. We have clarified the collection procedure and the diet procedure to address this point and others that the reviewer made. Blood samples were gathered at 8:00 pm (3 hours after the diet was placed on the cages). Using this time point, blood samples ranged from 14.90-110.70 mg/dL. A separate study using the same administration paradigm (Bodnar et al., 2021) gathered blood samples at 7:00 am and reported significantly higher BECs (122.2 ± 26 mg/dL). It is possible that we would have gotten higher BECs if blood samples were collected at the end of the dark cycle. We plan to adjust the timing of blood samples going forward to reflect this finding. We have added this information to the Materials and Methods. 

- We do have data for the amount of diet consumed for each dam on each day. On average, ethanol dams consumed 89.0 ± 8.24 mL of the liquid diet, which would equate to 5.96 mL of ethanol (95%) in a 24-hour period. We can add this data into the manuscript if requested.

• As mentioned previously, the PAE paradigm used is described in the Material and Methods section. The current version is not clear to readers, it would be better to add more details such as the age of female rats before starting with the paradigm, and the number of litters used. It seems that for each experimental group, the authors used one litter since it is reported in line 134 “ with the exception of the alcohol-fed litter where only two females were available”. Please clarify this point.

- The females used for breeding were approximately 9 months of age, which has been added to the manuscript for clarification. It is correct that for tissue collection, offspring were derived from one litter per treatment group. We elected to limit litter variability and include males and females in the analysis. We realize that this is a limitation of the current study and have addressed this point in the discussion (paragraph inserted below for ease).

- “As with any study, there are limitations to the current data. First, we elected evaluate gene expression in the hippocampus only, while acknowledging that gene expression, and its changes following PAE are likely to be region-dependent. However, changes in gene expression of protocadherins following developmental exposure have been observed in mouse cortex [12], human buccal samples [16,37], and zebrafish embryo [38], suggesting that this may be a reliable consequence of prenatal alcohol exposure. Another important factor to consider is the timing of alcohol exposure. In the rodent, prenatal alcohol administration results in exposure that is limited to the first and second trimester periods. It is entirely possible that a different outcome would be observed with postnatal exposure during the brain growth spurt (PD 4-9). Future studies should evaluate whether the timing of exposure is an important factor in the current outcome. Finally, we were limited in the number of samples that could be analyzed in the current study. After finding no sex differences, we elected to combine males and females, resulting in n’s of 5-6/group (2-3 males and females) Further, these rats were derived from a single litter per treatment group, with the goal of minimizing litter variability in order to focus on treatment effects. We recognize this as a limitation of the current work and plan to replicate and expand these studies going forward. However, the current findings align well with the existing literature, providing support for the current data. Finally, future studies should validate the expression changes observed in the current study..”

Also, the readers have to assume that the exposure to alcohol started during the first day of gestation (G1) until G20. Please add more details about the protocol. Any specific reason for choosing this period of time?

- Yes, diet administration occurred throughout gestation. Diet administration started on G1 and continued to G20, when all rats were switched to standard food and water. This timeframe was chosen to encompass the first- and second-trimester equivalent periods of brain development. This point has been clarified in the revised manuscript. 

• Furthermore, it would be more accurate to report the concentration of Ethanol used in the experimental diet and the average consumption during 1.5-2 hours of exposure- drinking in the dark. Furthermore, it is not clear if the dams were exposed for 1.5 or 2 hours. Clarify this critical point.

- This point has been clarified in the revised manuscript. This was not a drinking in the dark procedure. Instead, the liquid diet was available the entire time during gestation. Water was available ad libitum, but the liquid diet was the only source of food. This approach has been used before in the PAE field (e.g., Riley et al., 1979, Driscoll et al., 1980, Holman et al., 2018, Bodnar et al., 2021). See also Marquadt & Brigman (2016) for an excellent review of the behavioral consequences of developmental alcohol exposure in rodents that includes details about experimental paradigm. 

- We do have data for the amount of diet consumed for each dam on each day. On average, ethanol dams consumed 89.0 ± 8.24 mL of the liquid diet, which would equate to 5.96 mL of ethanol (95%) throughout the 24-hour exposure period. We can add this data into the manuscript if requested. 

• In the same section, authors reported the analysis performed for comparing possible differences in the body weight of pups at P7, showing a significant difference between males and females. Add more details about the statistical test used for those comparisons.

- Thank you for this suggestion and we apologize that the statistical analyses used were not described appropriately. We conducted an ANOVA to evaluate body weight at PD 14, 21, and at tissue collection and have added this information into the manuscript. Since we did not observe many differences, we chose to briefly mention this analysis and not focus on weight in the results section. 

• In the section “Hippocampus tissue collection”, the authors reported that the animals were decapitated without anesthesia. Why did they decide to use this procedure? The stress during the sacrifice could be affected the result. Do the authors consider the possible contamination of blood?

- Prenatal alcohol exposure can result in significant changes to the HPA axis. Thus, we wanted to collect the tissue as quickly as possible, to avoid possible differences as a function of stress exposure. Tissue collection occurred quickly (within 3-4 minutes) upon removing the rat from the home cage. We have added a sentence to clarify this point in the Materials and Methods section. 

- We cannot rule out the possible contamination of blood, and future studies may compare the current findings to gene expression in tissue that has been perfused with saline prior to collection. 

• In the discussion section, the authors compared their findings with the current literature talking about differences related to the brain region examined and sex differences. However, they did not mention the possible differences related to the prenatal alcohol exposure paradigm used (prenatal vs 3rd trimester), the alcohol concentration.

- This is an excellent point and we have added this to the discussion (Lines 415-432 of the revised manuscript)

Minor comments

1. The authors across the entire manuscript use the abbreviation FASD for Fetal Alcohol Spectrum Disorders as indicated in line 29, since they are using the plural of disorder, it would be more correct to add the “s” to the abbreviation FASD.

- The standard abbreviation for Fetal Alcohol Spectrum Disorders is FASD (example reference: Mattson et al., 2019, May et al., 2009). The full term is already in plural form, so adding an ‘s’ to the abbreviation would not be appropriate. 

2. Please indicate the acronym for Fetal Alcohol Syndrome (FAS).

- This has been corrected.

3. Several times, the authors did not spell out the acronyms used, for example, it would be more appropriate to define MRI (line 74), DTI (line 76).

- This has been corrected.

4. In the Author Summary section, it is used the term mothers for the rat dams. It would be more appropriate to use the term dams.

- This has been corrected.

5. In line 67, the authors defined the acronym PAE for Prenatal Alcohol Exposure, but they did not use it in the entire manuscript. Please be consistent in using the acronyms. Lines: 81, 90, 100, 106, 109, 111, 114, 119, 122, 129, 249, 268, 274, 278, 280, 282, 308, 311, 313, 315, 320.

- This has been corrected, with the exception that the full term still remains in subheadings and at the beginning of the introduction to remind readers of the abbreviation. 

6. When they are going to indicate the age of the animals, explain for the first time the meaning of P indicating postnatal day.

- This has been corrected.

7. Table 1: typo for “adhesion”

- This has been corrected.

Reviewer #2: In this manuscript, Amal Khalifa, et al. isolated RNA from the hippocampus of adolescent rats exposed to ethanol during prenatal development and compared gene expression to controls. They found many genes were downregulated and these genes were associated with cell adhesion, toxin metabolism, and immune responses. They concluded from this study that changes in genetic architecture after prenatal exposure to alcohol continue through adolescent development.

There are some concerns.

1. From fetus to adolescence, there are many factors that could affect the changes in gene expression. Even among littermates, gene expression can vary significantly. Therefore, in order to identify the effect of fetal alcohol exposure on adolescent gene expression, a relatively large sample size is required to eliminate the influence of these factors. In this study, three males and three females were sampled for each group, with the exception of the alcohol-fed group that only two females were sampled. When analysis, three outlier samples were removed. The results from a small sample size study are likely affected by many factors, resulting in false positive results.

- We recognize that this is a limitation of the current study and have added this to the discussion. We elected to include both males and females in the analysis, but few sex differences were observed. Thus, we were able to combine the males and females in the analysis. In addition, we realize that there is variability even within litters, so we chose to limit our analysis to a single litter for each group, in order to avoid introducing litter effects into the analysis. Furthermore, we have added some references throughout the paper that also found changes in expression of protocadherins following prenatal alcohol exposure. We feel that this helps provide support from the literature for the current data. We have clarified these important points throughout the manuscript. 

2. For the most differentially expressed genes, the expression and function of these genes of interest need to be further validated.

- We agree, and have included this as a point in the discussion. However, it is beyond the scope of the current manuscript. 

3. As for sex differences, in Table S5, the manuscript compared ET Males with ET Females. In Table S6, compared PF Males with PF Females. As readers and reviewers, we also hope to know the comparison between ET Males and PT Males and the comparison between ET Females and PT Females. I think the authors should perform the comparisons and include them into the manuscript, even though the principle components analysis (PCA) shows that sex is not a primary factor.

- We thank the reviewer for the suggestion to include this information in the manuscript. We have now included two additional supplemental tables (Tables S10-11) comparing PF and ET male samples and female samples independently. We also provide a visual comparison of these lists in Venn diagrams to demonstrate the overlap between down and upregulated genes in all three analyses (Supplemental Figure 2). These data demonstrate that most of the genes identified in sex-specific analysis were also identified in the analysis on all compiled samples. Furthermore, there are only a small number of genes uniquely identified in male or female-only analysis, suggesting that biological sex has little impact on differential gene expression in this study. Furthermore, we see a substantial increase in significantly altered genes when male and female samples are compiled, demonstrating the increased power we gain by increasing biological replicates in this way. We now include these explanations in the text. 

4. On page 25, in the first line, in Figure 2B, the volcano plot was generated from data comparing ET rats to AD rats, not PF rats. This might be a typo.

- We appreciate this comment. It was in fact a typo that has been corrected. 

6. PLOS authors have the option to publish the peer review history of their article (what does this mean?). If published, this will include your full peer review and any attached files.

Do you want your identity to be public for this peer review? For information about this choice, including consent withdrawal, please see our Privacy Policy.

Reviewer #1: No

Reviewer #2: No

---

## [Decision Letter · Decision Letter 1]

30 Aug 2023

PONE-D-23-18670R1Prenatal alcohol exposure alters expression of genes involved in cell adhesion, immune response, and toxin metabolism in adolescent rat hippocampusPLOS ONE

Dear Dr. Rebecca Palu,

Thank you for submitting your manuscript to PLOS ONE. After careful consideration, we feel that it has merit but does not fully meet PLOS ONE’s publication criteria as it currently stands. Therefore, we invite you to submit a revised version of the manuscript that addresses the points raised during the review process.

ACADEMIC EDITOR:

Thanks for the revision and responses. All reviewers evaluated that this manuscript as potential important. However, the findings are still hampered by several limitations. I believe that the comments from Reviewer #3 are constructive and addressing these comments will improve this manuscript.

We look forward to receiving your revised manuscript.

Kind regards,

Jia Luo, Ph.D.

Academic Editor

PLOS ONE

Reviewers' comments:

Reviewer's Responses to Questions

**Comments to the Author**

1. If the authors have adequately addressed your comments raised in a previous round of review and you feel that this manuscript is now acceptable for publication, you may indicate that here to bypass the “Comments to the Author” section, enter your conflict of interest statement in the “Confidential to Editor” section, and submit your "Accept" recommendation.

Reviewer #2: All comments have been addressed

Reviewer #3: (No Response)

2. Is the manuscript technically sound, and do the data support the conclusions?

Reviewer #2: Yes

Reviewer #3: Partly

3. Has the statistical analysis been performed appropriately and rigorously? 

Reviewer #2: Yes

Reviewer #3: Yes

4. Have the authors made all data underlying the findings in their manuscript fully available?

Reviewer #2: Yes

Reviewer #3: No

5. Is the manuscript presented in an intelligible fashion and written in standard English?

Reviewer #2: Yes

Reviewer #3: Yes

6. Review Comments to the Author

Reviewer #2: The authors have satisfactorily addressed all the comments and revised the manuscript. I have no further concerns and questions.

Reviewer #3: This is a potentially important paper that documents gene-expression level changes that persist into adolescence in the PAE rat hippocampus. Too little is known about such persistence at the molecular / cellular level and these findings are potentially an important contribution to the field. The use of an unbiased approach is another strength, as it can reveal novel targets.

However, the findings are hampered by several limitations, some due to a flawed study design, and others that can be improved in the data presentation. The authors should want their data to shine! These broad suggestions, which I hope are constructive, are followed by specific concerns.

1. This is a 3-way study design that compares Et, pair-fed, and ad lib controls. Thus, the most important gene-level changes are the 363 genes altered by Et and not PF+AD. The pathway analysis should focus on this data set (not Et-PF only) because these expression-level changes transcend diet and eating behavior (which is why presumably one runs both controls). This means repeating the DAVID (and I suggest adding KEGG, which is more informative than GO-terms), but the reanalysis should be swift and the results far more meaningful.

2. Acknowledge the study design details upfront in Methods; don’t bury them in Discussion. I had to dig to discover the experimental blocking. The progeny of only one dam per treatment was sampled; these are not true biological replicates and it is unknown whether the findings are representative. Thus, the study should be labeled as ‘exploratory’. In this reviewer’s experience, inter-litter variance in RNA-Seq is not excessive. Also, technical replicates are unnecessary as the technology is mature. It is a shame that the expense of high-throughput sequencing was expended on littermates from a single dam.

3. Similarly, because BAL was not quantified in the Et dam, her exposure is unknown and cannot be ‘guesstimated’ due to the model’s high variance (15 – 110 mg/dL). Note that only 10-20ul are needed for BAL, readily obtained from a tail nick and a capillary tube.

4. The most important findings are hidden in Supplemental Tables. As the key results are in the GO-term (and I suggest KEGG) analysis, these should be presented in the Results. Many of the supplemental tables could be condensed into a single one for Results, and extraneous data within them could be omitted. I suggest that this table should focus on the 363 common gene set, merge or select related GO/KEGG terms, and exclude those terms that were not significant in the P-adj.

Specific Comments:

Abstract – Take advantage of the 300-word limit and describe the Methods and Results more fully. The Abstract Introduction could be trimmed. Add details of alcohol exposure (timing, dose) and details of the pathway-level expression changes. The current abstract undersells the findings.

Introduction

Line 62 – Ref 1 is outdated; replace with most recent BFSS findings and report accurately in text (don’t round).

Lines 65-66 – FAS and FASD are described incorrectly. FASD is not a milder consumption. FAS is under the FASD umbrella. Refer to the Hoyme Diagnostics for a fuller understanding.

Lines 70-85 – Review of current hippocampal findings omits a number of key clinical and preclinical findings. Not sure why this section emphasizes an older literature.

Line 87 – PAE doesn’t alter the genome (=DNA) but rather gene-level expression, in part perhaps thru methylation differences. However, alterations in cell composition could also change expression results in a heterogeneous tissue such as hippocampus. Just be aware of this.

Methods

Line 127 – 9-mo old female rats are considered very old for breeding. Provide the rationale for their selection (am guessing they were free). Add results (to Results) regarding the impact of PAE and pair-feeding on maternal weight gain. It is unclear how PAE and PF might have affected maternal, fetal, and offspring growth. Please add this to Results; for example, did the PAE offspring experience catch-up growth?

Lines 131 and 148 – there is no such thing as “standard chow” – every rodent chow has a different composition. Please provide the diet # and vendor for the chow provision. The chow composition differs from that of Weinberg-Keiver diet and this is a Limitation. A better control for future studies would have been ad libitum of the liquid diet.

Line 144-146 – BAL values are a Result not a Method. Describe the method used to quantify BAL. Provide the units for that measurement. Add the mean quantity of alcohol consumed per day to Results. Given the huge variance in the BALs, the authors might show this variance in a scatterplot. Unfortunately, BALs were not made in the sequencing dam, so it is not possible to relate BAL to expression-level changes in the offspring; this is a Limitation. Long-Evans rats are outbred – consider prescreening dams for those who are drinkers to generate more consistency in BALs, or rethink when BAL is sampled.

Lines 150-157 – do not include Results in the Methods.

Line 159 – critical details regarding experimental blocking are missing. How many dams per group? How many offspring per group? How many offspring per litter were sampled, and do the results need to be blocked by litter? Were the sampled offspring cagemates if they came from the same dam? We do not learn until the Discussion that apparently only one dam per treatment was used – this is a huge limitation, especially given the high variance in the BALs, which were not measured in the PAE dam who was actually used. Sampling individual animals from multiple litters is actually a strength, because it transcends variance and reveals the consistent outcomes. The current study design cannot distinguish interlitter variance and indeed discarded one such animal for ‘cleaner data’ – this is not good practice.

Line 160-161 – decapitation is very unusual in rodent studies and was probably unnecessary here; gene expression doesn’t change that rapidly under the 2-3min of isofluorane. Information that the animal protocols were approved by the Institutional Animal Use Committee should be at line 126 not lines 229-230. State that the Purdue facility is AAALAC accredited (I assume so).

Line 178 – how long were the paired-end reads? The verb tenses in this paragraph are odd and read like they were copied from facility site. Include experimental blocking details – how many biological replicates per group? How many technical replicates? Were results blocked by litter? There were apparently 51 sequencing runs (line 178) but only 2-3 rats per treatment per sex (line 160, 3 rats x 3 txmt x 2 sexes = 18 biological replicates?). Three technical replicates per sample would be 54 samples. What influenced the different numbers per group?

Lines 204-207 – is this information necessary?

Results

Read carefully and remove all Methods from the Results. Experimental blocking (lines 236-240) belongs in Methods. Results begin on Line 244. Similarly, remove all Discussion from Results (for example, lines 353-364).

I did not find a statement that described where the data will be deposited.

The first paragraph in Results should include the maternal and offspring growth characteristics, alcohol consumption levels, and the BAL data. This information defines your model and would be cited in future papers that use it.

Figure 3 Legend. Remove key results and present them in the Results (lines 341-345).

Line 234-235 is misleading as suggests that multiple PAE dams contributed to the study. In truth, the authors have no idea if this single dam experienced 15 or 110 mg/dL. This should be directly acknowledged in Limitations; transparency is key and makes the data look better, not worse.

Line 242-243 – it is not true that the liquid diets are identical except for ethanol – the control diet substitutes carbohydrate calories for the alcohol calories, and these are metabolized differently. Be aware of this as it can influence outcomes; even its study designers are aware of this.

Figure 1B – it is odd that PC1 doesn’t separate the Et and PF samples, nor does PC2. The Discussion should comment on this.

Line 299 – As only 7 genes were consistent in all three comparisons of Et genes, those could be listed in the text so readers don’t have to sort through the supplemental tables.

Lines 308-319 – The key result is the identification of 347 decr and 16 incr genes common to the comparisons of Et-PF and Et-AD. This should be the focus of Results and Discussion rather than Et-PF because these results transcend diet. The other Results up to this point could be trimmed.

Figure 2 Legend and Line 315 – clarify the criteria for “Top 10”– is it Padj or Fold-Change? The volcano plot could be moved to Supplemental. This would free space for more relevant data tables otherwise relegated to Supplement (see below).

Figure 3 Legend. Define the X- and Y-axes. Is Y-axis Padj or FC? Annotate X-axis with respect to which samples are Male and Female. These genes are presumably from the Common Set (N=363); why then do only 7/10 have significant change compared to PF and AD? Remove results from Legend to Results Text.

Lines 368 forward – why is the emphasis on DAVID instead of KEGG analysis? KEGG is a more powerful tool to identify altered pathways. Consider running DAVID and KEGG on the entire Common Set (up + down combined); remember both down- and up-regulated genes can contribute to the same pathway. This approach would better leverage the few increased genes.

Table 1 and Lines 379-384. The GO numbers are not particularly meaningful as numbers. What is more meaningful in such a table is the GO term name, Padj, list of gene names in that category (could merge esp for those with overlapping members), and the number of hits in that category vs. possible members that comprise the term. This is probably easier and more meaningful as KEGG instead of GO. This list should be a Table and not relegated to Supplemental (S12-26).

Figure 4 does not show the GO findings clearly (see below) – it could be Supplemental and instead present as a Table. Label Y-axis. Again, if these are supposed to be significant differences, why are only some significantly different (marked by *). Again, focus on the Common Set of 363.

Lines 407-424 again are largely Discussion, not Results. Having a Table that lists the genes in these broad categories would greatly enhance the manuscript.

The presence of 26 Supplemental Tables does not serve the data or manuscript well. Move important findings to the Results. Specifically, Tables S12 – S26 are the ‘meat’ of the paper. Look at these and pull forward into a Results Table the most important GO findings from these – Cell Adhesion, Glucuronidation, Immunity, Toxin Metabolism. Merge overlapping GO terms into a single gene list. Much of the information in those tables is extraneous – focus on the Gene List, count, FC, P-value, and Bonferroni OR Benjamini OR FDR (note Benjamini is often misspelled in these supplemental tables). The Supplement can contain the less significant GO terms (and many of those tables could be merged for ease of reading). GO terms that are not significant under the FDRs should not be reported, because they are not significant (i.e. S26, S24, and data in other tables). Much of the information in S12-S26 is distracting.

It is unclear why Tables S12-S26 focus on Et-PF and not the Common Set of Et-PF+AD. Those are the most interesting and meaningful genes, as they respond to Et and transcend diet.

The image quality of Figures 1-4 is poor. Images at 72dpi can be saved as .png format, then expanded to 300dpi and resized without loss of resolution. Photoshop does this nicely.

Table 1 – why is this an image instead of a text file?

Reviewer 2 suggested replication of expression using, say qPCR. This is not standard in whole transcriptome work because the method uses direct sequencing of cDNA, and thus provides near-absolute copy number. However, replication would be useful due to the lack of true biological replicates, so such should be noted in Limitations. Note I am not asking for replication here, as the Results should be described as ‘exploratory’.

Discussion

Remove Discussion items from Results.

Limitations should be its own paragraph. Such actually strengthens the paper as it provides transparency.

7. PLOS authors have the option to publish the peer review history of their article (what does this mean?). If published, this will include your full peer review and any attached files.

Reviewer #2: No

Reviewer #3: No

---

## [Author Response · Author response to Decision Letter 1]

22 Sep 2023

PONE-D-23-18670R1

Response to Reviewers

 We thank the reviewers for their insightful comments and have provided a point-by-point response below. 

Comments to the Author

1. If the authors have adequately addressed your comments raised in a previous round of review and you feel that this manuscript is now acceptable for publication, you may indicate that here to bypass the “Comments to the Author” section, enter your conflict of interest statement in the “Confidential to Editor” section, and submit your "Accept" recommendation.

Reviewer #2: All comments have been addressed

Reviewer #3: (No Response)

2. Is the manuscript technically sound, and do the data support the conclusions?

Reviewer #2: Yes

Reviewer #3: Partly

3. Has the statistical analysis been performed appropriately and rigorously?

Reviewer #2: Yes

Reviewer #3: Yes

4. Have the authors made all data underlying the findings in their manuscript fully available?

Reviewer #2: Yes

Reviewer #3: No

5. Is the manuscript presented in an intelligible fashion and written in standard English?

Reviewer #2: Yes

Reviewer #3: Yes

Review Comments to the Author

Reviewer #2: The authors have satisfactorily addressed all the comments and revised the manuscript. I have no further concerns and questions.

Reviewer #3: This is a potentially important paper that documents gene-expression level changes that persist into adolescence in the PAE rat hippocampus. Too little is known about such persistence at the molecular / cellular level and these findings are potentially an important contribution to the field. The use of an unbiased approach is another strength, as it can reveal novel targets.

However, the findings are hampered by several limitations, some due to a flawed study design, and others that can be improved in the data presentation. The authors should want their data to shine! These broad suggestions, which I hope are constructive, are followed by specific concerns.

- We thank the reviewer for this comment and the contributions below. We completely agree and acknowledge some of the limitations of the current study but feel that the scarcity of available data on this topic warrants publication. We appreciate that the reviewer acknowledged this point and will emphasize it in the manuscript. The suggestions of the reviewer are addressed below.

1. This is a 3-way study design that compares Et, pair-fed, and ad lib controls. Thus, the most important gene-level changes are the 363 genes altered by Et and not PF+AD. The pathway analysis should focus on this data set (not Et-PF only) because these expression-level changes transcend diet and eating behavior (which is why presumably one runs both controls). This means repeating the DAVID (and I suggest adding KEGG, which is more informative than GO-terms), but the reanalysis should be swift and the results far more meaningful.

- We thank the reviewer for this helpful suggestion. We initially used only ET versus PF to maximize the number of genes included in the GO analysis. We repeated the analysis using only those genes that are altered in ET compared to both PF and AD controls. As expected, we identified the same pathways and processes using this smaller subset of genes. In particular, our analysis highlighted cell adhesion and glucoronidation as particularly of interest. While the reduced number of genes reduced the number of significant categories (as we predicted) we found several highly significant categories for each of the processes highlighted in Table 2. Part of this analysis also included KEGG pathway analysis (Table S19), which did highlight a number of interesting pathways in drug and toxin metabolism as well as glucoronidation once again. 

2. Acknowledge the study design details upfront in Methods; don’t bury them in Discussion. I had to dig to discover the experimental blocking. The progeny of only one dam per treatment was sampled; these are not true biological replicates and it is unknown whether the findings are representative. Thus, the study should be labeled as ‘exploratory’. In this reviewer’s experience, inter-litter variance in RNA-Seq is not excessive. Also, technical replicates are unnecessary as the technology is mature. It is a shame that the expense of high-throughput sequencing was expended on littermates from a single dam.

- We have edited the methods to reflect the reviewer’s suggestion. 

- We acknowledge that sampling from a single litter per treatment group is a limitation. Unfortunately, breeding was a major issue at the time that this study was completed. While we had planned to use no more than 1-2 rats per litter per group, only one ethanol-exposed dam gave birth during the study, and we were on a deadline to complete the study while funds were available. We absolutely accept this as a limitation and address it in the paper. 

3. Similarly, because BAL was not quantified in the Et dam, her exposure is unknown and cannot be ‘guesstimated’ due to the model’s high variance (15 – 110 mg/dL). Note that only 10-20ul are needed for BAL, readily obtained from a tail nick and a capillary tube.

- Blood samples were gathered from this and other ethanol dams over the course of a year using the same model. Due to a single litter, we felt that it was more relevant to discuss a larger sample size for the BALs. In addition, the BAL for this dam was on the low end (16.9 mg/dL) but this dam, and all dams, consumed almost the entire bottle of diet each night. We are in the process of collecting samples at the end of the drinking bout, when lights go on, but do not have the data at this time. We have elected to discuss this point and include the amount of diet consumed for this dam (average = 88 mL, range = 63 mL on the first day, 100 mL on other days). If calculated using an average of 88 mL of diet, consumption would be 5.9 mL of 95% ethanol over the course of the drinking bout. This comes out to approximately 13 g of ethanol per kg of body weight. As such, it is very likely that the BAL value of 16.9 mg/dL is not representative of 

the peak BAL reached by this rat (and other rats) using this model. 

4. The most important findings are hidden in Supplemental Tables. As the key results are in the GO-term (and I suggest KEGG) analysis, these should be presented in the Results. Many of the supplemental tables could be condensed into a single one for Results, and extraneous data within them could be omitted. I suggest that this table should focus on the 363 common gene set, merge or select related GO/KEGG terms, and exclude those terms that were not significant in the P-adj.

- We agree that it is important to highlight specific GO terms results from our overall analysis, rather than just including the supplemental tables. We would direct the reviewer to Table 2, where we have done just that. This table was included as a separate file in the original submission, rather than embedded in the text as it should have been. We have corrected the error with apologies. We have also expanded the information in this table so that the audience can easily access the results that inform our primary conclusions and interpretations. We have chosen to retain the full set of results as supplemental tables. We acknowledge this makes our supplement somewhat expansive, but believe it is important to include all of our results as this is primarily an exploratory study. Should someone wish to follow up on results we did not highlight, we wish them to have the full set of information and results from which to work. 

Specific Comments:

5. Abstract – Take advantage of the 300-word limit and describe the Methods and Results more fully. The Abstract Introduction could be trimmed. Add details of alcohol exposure (timing, dose) and details of the pathway-level expression changes. The current abstract undersells the findings.

- We have revised the abstract to include suggestions by the reviewer. 

6. Introduction

Line 62 – Ref 1 is outdated; replace with most recent BFSS findings and report accurately in text (don’t round).

- We have updated this reference to include data from Gosdin et al. (2022). 

7. Lines 65-66 – FAS and FASD are described incorrectly. FASD is not a milder consumption. FAS is under the FASD umbrella. Refer to the Hoyme Diagnostics for a fuller understanding.

- The reviewer is correct that the terms were used incorrectly. This statement has been changed. 

8. Lines 70-85 – Review of current hippocampal findings omits a number of key clinical and preclinical findings. Not sure why this section emphasizes an older literature.

- We thank the reviewer for this helpful comment. We have reviewed and updated some of the literature referenced in this paragraph, while still focusing the introduction primarily on studies evaluating epigenetic and genetic changes following prenatal alcohol exposure. 

9. Line 87 – PAE doesn’t alter the genome (=DNA) but rather gene-level expression, in part perhaps thru methylation differences. However, alterations in cell composition could also change expression results in a heterogeneous tissue such as hippocampus. Just be aware of this.

- Yes, we agree that this term could have been more precise and have adjusted it. 

Methods

10. Line 127 – 9-mo old female rats are considered very old for breeding. Provide the rationale for their selection (am guessing they were free). Add results (to Results) regarding the impact of PAE and pair-feeding on maternal weight gain. It is unclear how PAE and PF might have affected maternal, fetal, and offspring growth. Please add this to Results; for example, did the PAE offspring experience catch-up growth?

- This information has been added to the results. In addition, maternal and offspring characteristics are now included in Table 1. While we acknowledge that 9 months is on the higher end for breeding age, it is within the age considered “adult”, with reproductive senescence beginning around 10 months of age (Cruz et al., 2017). We have noticed a significant decline in the number of successful litters (pups born following presence of a sperm plug) after 10 months. 

11. Lines 131 and 148 – there is no such thing as “standard chow” – every rodent chow has a different composition. Please provide the diet # and vendor for the chow provision. The chow composition differs from that of Weinberg-Keiver diet and this is a Limitation. A better control for future studies would have been ad libitum of the liquid diet.

- This is a great point. We have now included the manufacturer and diet # for the rat chow. The ad lib control diet does differ from the Weinberg-Keiver diet and this is acknowledged as a limitation. In fact, in recent publications from the Weinberg lab, control rats are given ad libitum access to a pelleted version of the control diet (e.g., Bodnar et al., 2021; Holman et al., 2021). This strategy avoids the stress associated with immediate consumption of the liquid diet in pair-fed rats. We appreciate this comment. 

12. Line 144-146 – BAL values are a Result not a Method. Describe the method used to quantify BAL. Provide the units for that measurement. Add the mean quantity of alcohol consumed per day to Results. Given the huge variance in the BALs, the authors might show this variance in a scatterplot. Unfortunately, BALs were not made in the sequencing dam, so it is not possible to relate BAL to expression-level changes in the offspring; this is a Limitation. Long-Evans rats are outbred – consider prescreening dams for those who are drinkers to generate more consistency in BALs, or rethink when BAL is sampled.

- This is a great point. So far, we have had only less than 5% of dams removed from the studies for failing to consume the experimental diet. This is typically not an issue for us, but we do note the low BALs acquired from this dam (see comment above). We are confident that it is not an issue of failing to consume the diet (average of 88mL of liquid diet per day for this rat) and instead think it is an issue of when the blood samples were collected, as the reviewer pointed out. We have recently decided to switch the time of blood collection to 7am (+12 hours after diet is placed on the cage) when the lights go on. Unfortunately, we do not have the data yet but really value this comment. 

- The BALs were assessed using an Analox machine and this information has now been added to the methods. 

13. Lines 150-157 – do not include Results in the Methods.

- We have reorganized the order of material presented to be consistent with this suggestion.

14. Line 159 – critical details regarding experimental blocking are missing. How many dams per group? How many offspring per group? How many offspring per litter were sampled, and do the results need to be blocked by litter? Were the sampled offspring cagemates if they came from the same dam? We do not learn until the Discussion that apparently only one dam per treatment was used – this is a huge limitation, especially given the high variance in the BALs, which were not measured in the PAE dam who was actually used. Sampling individual animals from multiple litters is actually a strength, because it transcends variance and reveals the consistent outcomes. The current study design cannot distinguish interlitter variance and indeed discarded one such animal for ‘cleaner data’ – this is not good practice.

 - We acknowledge this as a limitation of the current study and, as explained above, originally intended to sample from multiple litters for each group. Due to issues in breeding at the time, only one ethanol dam that was bred gave birth. . 

15. Line 160-161 – decapitation is very unusual in rodent studies and was probably unnecessary here; gene expression doesn’t change that rapidly under the 2-3min of isofluorane. Information that the animal protocols were approved by the Institutional Animal Use Committee should be at line 126 not lines 229-230. State that the Purdue facility is AAALAC accredited (I assume so).

- We must respectfully disagree that decapitation is very unusual in rodent studies. In addition, decapitation without anesthesia is also not uncommon (see Bodnar et al., 2016; Lam et al., 2018; Ruffaner-Hanson et al., 2023; Marsland et al., 2023). While we acknowledge that it is possible to repeat the study by using isoflurane prior to decapitation, and may do so going forward, we were partly concerned by the potential confound of the physiological stress response. Prenatal alcohol exposure has been shown to differentially alter the physiological stress response (see Ruffaner-Hanson et al., 2023, Lam et al., 2018). However, we value this comment and are now reviewing the literature regarding isoflurane anesthesia and the physiological stress response (see Zardooz et al., 2010, Wu et al., 2015, Vahl et al. 2005). 

- We have added the information about AAALAC accreditation to the Methods section under “Animals”. 

16. Line 178 – how long were the paired-end reads? The verb tenses in this paragraph are odd and read like they were copied from facility site. Include experimental blocking details – how many biological replicates per group? How many technical replicates? Were results blocked by litter? There were apparently 51 sequencing runs (line 178) but only 2-3 rats per treatment per sex (line 160, 3 rats x 3 txmt x 2 sexes = 18 biological replicates?). Three technical replicates per sample would be 54 samples. What influenced the different numbers per group?

- Thank you for the suggestions. We have reviewed and updated this section to include more detailed information on the sequenced samples. Since the exact length of each sequenced sample is different, we provided an estimated average value for the number of reads per file. We have reported the number of biological replicates in each group with details on their sex as well. The text now has a clear explanation of the total number of sequences used in the analysis (17 samples x 3 runs = 51).

17. Lines 204-207 – is this information necessary?

- We feel this information is important for the methods section of the paper. It highlights the steps taken during taken to analyze the aligned reads to look for differential expression. Including this type of methodological information ensures anyone wishing to replicate our study will have the full information about what tools and what programs we used to obtain the results reported here. 

Results

18. Read carefully and remove all Methods from the Results. Experimental blocking (lines 236-240) belongs in Methods. Results begin on Line 244. Similarly, remove all Discussion from Results (for example, lines 353-364).

- We have reorganized the manuscript as suggested by the reviewer. 

19. I did not find a statement that described where the data will be deposited.

- We will deposit the data in the NCBI GEO database (https://www.ncbi.nlm.nih.gov/geo/)

20. The first paragraph in Results should include the maternal and offspring growth characteristics, alcohol consumption levels, and the BAL data. This information defines your model and would be cited in future papers that use it.

- We have added and/or moved maternal and offspring data in the Results section of the manuscript. 

21. Figure 3 Legend. Remove key results and present them in the Results (lines 341-345).

- We have adjusted both legend and results text so that details are present in the results.

22. Line 234-235 is misleading as suggests that multiple PAE dams contributed to the study. In truth, the authors have no idea if this single dam experienced 15 or 110 mg/dL. This should be directly acknowledged in Limitations; transparency is key and makes the data look better, not worse.

- We have clarified this point in the manuscript. In addition, we have altered the paragraph about limitations to emphasize the exploratory nature of this study. 

23. Line 242-243 – it is not true that the liquid diets are identical except for ethanol – the control diet substitutes carbohydrate calories for the alcohol calories, and these are metabolized differently. Be aware of this as it can influence outcomes; even its study designers are aware of this.

- Yes, the reviewer is correct that the diets are not identical. More information regarding the diets has been added to the Methods section. 

24. Figure 1B – it is odd that PC1 doesn’t separate the Et and PF samples, nor does PC2. The Discussion should comment on this.

-We thank the reviewer for noting this. While the individual PC do not separate the two groups, we would point out the in PC analysis, we are looking for individual samples of distinct groups to cluster together and separately from the other group. While each individual PC does not do this on its own, we highlight in these figures that the two groups do cluster separately from one another, indicating that together the PC1 and PC2 do separate the groups. We note this now in the results and discussion. 

25. Line 299 – As only 7 genes were consistent in all three comparisons of Et genes, those could be listed in the text so readers don’t have to sort through the supplemental tables.

-We have made this addition.

26. Lines 308-319 – The key result is the identification of 347 decr and 16 incr genes common to the comparisons of Et-PF and Et-AD. This should be the focus of Results and Discussion rather than Et-PF because these results transcend diet. The other Results up to this point could be trimmed.

-We have updated our GO and pathway analysis to focus on the combined analysis of genes differentially expressed as compared to PF and AD controls. We do feel it is important to note in the top genes those that are unique to PF and AD comparisons, which is why the data is included in Figure 3 and discussed as such in the results. Ultimately, our conclusions are the same whether we focus only on the ET versus PF comparison or focus on the genes that are changed in ET rats as compared to PF or AD. 

27. Figure 2 Legend and Line 315 – clarify the criteria for “Top 10”– is it Padj or Fold-Change? The volcano plot could be moved to Supplemental. This would free space for more relevant data tables otherwise relegated to Supplement (see below).

-We apologize for the confusion: this choice was made based on Padj (most significant) rather than Fold Change (biggest change). We have clarified this in both the results text and the Figure 3 legend. We feel this plot is important to show the degree of differential gene expression, and would prefer to leave this figure as part of the main text. As we only have four primary figures and 1 primary table, we do not feel space is unduly taken by inclusion of this figure. 

28. Figure 3 Legend. Define the X- and Y-axes. Is Y-axis Padj or FC? Annotate X-axis with respect to which samples are Male and Female. These genes are presumably from the Common Set (N=363); why then do only 7/10 have significant change compared to PF and AD? Remove results from Legend to Results Text.

- Figure 3 is a heat map. As indicated in the legend, biological samples are grouped along the X-axis, while genes are shown on the y-axis. P-value is indicated by the color of the square as indicated by the legend. As indicated in the text, these top ten genes came from the PF versus ET list. While the majority of the genes are shared, the order is not the same for AD versus PF. We wish here to highlight the degree of concordance of expression of samples in ET as compared to either PF or AD. We indicate the genes that are not significantly changed in AD samples with asterisks, which is described in the legend. We have adjusted the text to ensure primary results are discussed in more detail in the results text rather than in the legend. 

Male and female samples are now marked along the X-axis. 

29. Lines 368 forward – why is the emphasis on DAVID instead of KEGG analysis? KEGG is a more powerful tool to identify altered pathways. Consider running DAVID and KEGG on the entire Common Set (up + down combined); remember both down- and up-regulated genes can contribute to the same pathway. This approach would better leverage the few increased genes.

-We thank the reviewer for this suggestion. We wish to point out that part of the reason we selected DAVID for our GO analysis is that KEGG is included. These results are part of the analysis (Table S19) and this is highlighted in our revised Table 2. 

30. Table 1 and Lines 379-384. The GO numbers are not particularly meaningful as numbers. What is more meaningful in such a table is the GO term name, Padj, list of gene names in that category (could merge esp for those with overlapping members), and the number of hits in that category vs. possible members that comprise the term. This is probably easier and more meaningful as KEGG instead of GO. This list should be a Table and not relegated to Supplemental (S12-26).

-We appreciate the reviewer’s concern about the information in the supplemental material. We agree it is important to highlight the primary processes that were emphasized by the GO results and KEGG pathway analysis. This is why in Table 2, we break down the GO and KEGG analysis by broader category and highlighted the terms that fit into those categories from the larger analysis. We reserve the bulk of the technical information for the supplemental tables. This enables the readers to see the results in their entirety, and for them to draw their own conclusions should they wish to expand upon our results based on their own interpretation rather than our own. We have expanded Table 1 to include more information about each category we believe to highlight the processes in general. 

21. Figure 4 does not show the GO findings clearly (see below) – it could be Supplemental and instead present as a Table. Label Y-axis. Again, if these are supposed to be significant differences, why are only some significantly different (marked by *). Again, focus on the Common Set of 363.

-This information has been updated to include the genes and categories highlighted by the new Table 2 that was generated from the differentially expressed genes that appear in comparison to both PF and AD control groups. This change has resolved the concern about only some genes being significantly changed, since those marked with an asterisk were significantly changed compared only to PF and not AD controls. We like this figure as a visual similar to Figure 3 showing that the genes in question that highlight interesting categories for follow-up are in fact consistently changed in all analyzed samples, and have therefore kept it in its modified form as a main figure. 

22. Lines 407-424 again are largely Discussion, not Results. Having a Table that lists the genes in these broad categories would greatly enhance the manuscript.

-We have updated Table 2 to include more information about these categories. The genes themselves are highlighted in a more visual way in Figure 4. We felt in the results text itself that it was more important to highlight the broader categories and pathways as opposed to individual genes, as these processes are the true modifiers and regulators of the changes we observe in response to ethanol exposure. We have moved some of the description to the discussion, but retain some in the results to provide context for why we believe the relevant categories are in fact interesting and important in the response to prenatal alcohol exposure. 

23. The presence of 26 Supplemental Tables does not serve the data or manuscript well. Move important findings to the Results. Specifically, Tables S12 – S26 are the ‘meat’ of the paper. Look at these and pull forward into a Results Table the most important GO findings from these – Cell Adhesion, Glucuronidation, Immunity, Toxin Metabolism. Merge overlapping GO terms into a single gene list. Much of the information in those tables is extraneous – focus on the Gene List, count, FC, P-value, and Bonferroni OR Benjamini OR FDR (note Benjamini is often misspelled in these supplemental tables). The Supplement can contain the less significant GO terms (and many of those tables could be merged for ease of reading). GO terms that are not significant under the FDRs should not be reported, because they are not significant (i.e. S26, S24, and data in other tables). Much of the information in S12-S26 is distracting.

-While we appreciate the reviewer’s point of view, we disagree that the inclusion of these supplementary tables is unnecessary. In fact, we believe that their inclusion is particularly important since our study is exploratory. We feel that including the full results of the GO and pathway analysis ensures that readers wishing to dive deeper into the data are able to do so. It also ensures that in the future, anyone wishing to follow up on our work may be able to do so and make their own interpretation of our results as a whole, rather than relying only on the results we choose to include in the main text. We therefore are leaving the supplemental tables, while also expanding the information in Table 2 as requested so that the audience will have a more palatable source for the information on which we are basing our conclusions. 

24. It is unclear why Tables S12-S26 focus on Et-PF and not the Common Set of Et-PF+AD. Those are the most interesting and meaningful genes, as they respond to Et and transcend diet.

-We have updated these tables to include the appropriate information about re-doing the analysis.

25. The image quality of Figures 1-4 is poor. Images at 72dpi can be saved as .png format, then expanded to 300dpi and resized without loss of resolution. Photoshop does this nicely.

- We have updated the figures and hope they are of acceptable resolution.

26. Table 1 – why is this an image instead of a text file?

-We apologize for this oversight and have embedded this table in the text. We also include this as a separate .txt file, as the size of the table is somewhat unwieldy in the word document for the final manuscript. 

27. Reviewer 2 suggested replication of expression using, say qPCR. This is not standard in whole transcriptome work because the method uses direct sequencing of cDNA, and thus provides near-absolute copy number. However, replication would be useful due to the lack of true biological replicates, so such should be noted in Limitations. Note I am not asking for replication here, as the Results should be described as ‘exploratory’.

-While we agree this would be nice, this is beyond the scope of this study at this time. This is the focus of a future project for which we hope to prepare soon! However, the implementation of this is outside of the timeline of this publication. 

Discussion

28. Remove Discussion items from Results.

- We have reorganized the paper and removed discussion items from the results, when necessary. In some cases, we elected to keep the information since it provides context to the results. Specifically, providing a background for the key pathways highlighted by the analysis helps the reader understand why these pathways are important. 

29. Limitations should be its own paragraph. Such actually strengthens the paper as it provides transparency.

- We agree and want to point out that the second paragraph from the end of the discussion is a separate discussion of the limitations of the current study. 

7. PLOS authors have the option to publish the peer review history of their article (what does this mean?). If published, this will include your full peer review and any attached files.

Do you want your identity to be public for this peer review? For information about this choice, including consent withdrawal, please see our Privacy Policy.

Reviewer #2: No

Reviewer #3: No

---

## [Decision Letter · Decision Letter 2]

12 Oct 2023

Prenatal alcohol exposure alters expression of genes involved in cell adhesion, immune response, and toxin metabolism in adolescent rat hippocampus

PONE-D-23-18670R2

Dear Dr. Rebecca Palu,

We’re pleased to inform you that your manuscript has been judged scientifically suitable for publication and will be formally accepted for publication once it meets all outstanding technical requirements.

Kind regards,

Jia Luo, Ph.D.

Academic Editor

PLOS ONE

Additional Editor Comments (optional):

Reviewers' comments:

Reviewer's Responses to Questions

**Comments to the Author**

1. If the authors have adequately addressed your comments raised in a previous round of review and you feel that this manuscript is now acceptable for publication, you may indicate that here to bypass the “Comments to the Author” section, enter your conflict of interest statement in the “Confidential to Editor” section, and submit your "Accept" recommendation.

Reviewer #3: All comments have been addressed

2. Is the manuscript technically sound, and do the data support the conclusions?

Reviewer #3: Yes

3. Has the statistical analysis been performed appropriately and rigorously? 

Reviewer #3: Yes

4. Have the authors made all data underlying the findings in their manuscript fully available?

Reviewer #3: Yes

5. Is the manuscript presented in an intelligible fashion and written in standard English?

Reviewer #3: Yes

6. Review Comments to the Author

Reviewer #3: (No Response)

7. PLOS authors have the option to publish the peer review history of their article (what does this mean?). If published, this will include your full peer review and any attached files.

Reviewer #3: No

---

## [Editor Report · Acceptance letter]

11 Dec 2023

PONE-D-23-18670R2 

Prenatal alcohol exposure alters expression of genes involved in cell adhesion, immune response, and toxin metabolism in adolescent rat hippocampus 

Dear Dr. Palu:

I'm pleased to inform you that your manuscript has been deemed suitable for publication in PLOS ONE. Congratulations! Your manuscript is now with our production department. 

Kind regards, 

on behalf of

Dr Jia Luo 

Academic Editor

PLOS ONE